# Fine-tuned repression of Drp1-driven mitochondrial fission primes a 'stem/progenitor-like state' to support neoplastic transformation

Brian Spurlock[1], Danitra Parker[1], Malay Kumar Basu[2], Anita Hjelmeland[3], Sajina GC[3], Shanrun Liu[1], Gene P Siegal[4], Alan Gunter[1], Aida Moran[1], Kasturi Mitra[1]*

[1]Department of Genetics, University of Alabama at Birmingham, Birmingham, United States; [2]Departments of Pathology, University of Alabama at Birmingham, Birmingham, United States; [3]Department of Cell Development and Integrative Biology, University of Alabama at Birmingham, Birmingham, United States; [4]Departments of Pathology, Surgery, Genetics and Cell and Developmental Biology, University of Alabama at Birmingham, Birmingham, United States

**ABSTRACT:** Gene knockout of the master regulator of mitochondrial fission, Drp1, prevents neoplastic transformation. Also, mitochondrial fission and its opposing process of mitochondrial fusion are emerging as crucial regulators of stemness. Intriguingly, stem/progenitor cells maintaining repressed mitochondrial fission are primed for self-renewal and proliferation. Using our newly derived carcinogen transformed human cell model, we demonstrate that fine-tuned Drp1 repression primes a slow cycling 'stem/progenitor-like state', which is characterized by small networks of fused mitochondria and a gene-expression profile with elevated functional stem/progenitor markers (Krt15, Sox2 etc) and their regulators (Cyclin E). Fine tuning Drp1 protein by reducing its activating phosphorylation sustains the neoplastic stem/progenitor cell markers. Whereas, fine-tuned reduction of Drp1 protein maintains the characteristic mitochondrial shape and gene-expression of the primed 'stem/progenitor-like state' to accelerate neoplastic transformation, and more complete reduction of Drp1 protein prevents it. Therefore, our data highlights a 'goldilocks' level of Drp1 repression supporting stem/progenitor state dependent neoplastic transformation.

*For correspondence: kasturi@uab.edu

Competing interest: The authors declare that no competing interests exist.

## Introduction

The plasticity of the state of stemness is modulated by various intrinsic and extrinsic factors (*Magee et al., 2012*; *Folmes et al., 2012*; ). Involvement of mitochondria in the regulation of the stem cell state is complex (*Khacho and Slack, 2017*; *Lisowski et al., 2018*; *Zhang et al., 2018*). The master regulators of mitochondrial fission and fusion processes are emerging as crucial regulators of both embryonic and adult stem cells (*Khacho and Slack, 2017*; *Lisowski et al., 2018*; *Spurlock et al., 2020*). The Dynamin Related Protein 1 (Drp1) is the master regulator of mitochondrial fission that breaks larger mitochondria into smaller elements (*Friedman and Nunnari, 2014*; *Kageyama et al., 2011*). The effects of Drp1-driven mitochondrial fission is opposed by fusion between mitochondria driven by the Mitofusins and Opa1 (*Chen and Chan, 2017*; *Hoppins, 2014*; *Schrepfer and Scorrano, 2016*). Therefore, Drp1 repression allows unopposed mitochondrial fusion, and complete Drp1 repression sustains a hyperfused mitochondrial state. Enhanced mitochondrial fusion sustains stemness of certain adult stem cells (*Khacho et al., 2016*; *Senos Demarco et al., 2019*; *Wu et al., 2019*).

Particularly, repression of Drp1 supports stemness and repression of Mitofusin or Opa1 inactivates stemness in adult mouse neural lineage (*Iwata et al., 2020*; *Khacho and Slack, 2017*). The other extreme state, that is unopposed mitochondrial fission, is critical for achieving pluripotency of stem cells during reprogramming (*Prieto et al., 2016*). However, sustained mitochondrial fission reduces pluripotency of stem cells (*Zhong et al., 2019*). Thus, we speculate that a balance of timely fission and fusion of mitochondria may be critical for maintaining stem cell states.

In various tumors, the bulk tumor cell populations are maintained by the adult neoplastic stem cells (also called tumor initiating cells) as they self-renew, proliferate and differentiate (*Magee et al., 2012*). Drp1 has been linked to tumor formation in various cancer types (*Nagdas and Kashatus, 2019*; *Serasinghe et al., 2015*; *Tanwar et al., 2016*; *Tsuyoshi et al., 2020*; *Xie et al., 2015*). Drp1 activation sustains neoplastic stem cells at least in the astrocytic lineage (*Xie et al., 2015*), and genetic ablation of Drp1 prevents neoplastic transformation (*Serasinghe et al., 2015*). In contrast, an elevated mitochondrial fusion state, sustained by mitofusin, can drive immortalization of neoplastic stem cells to support tumorigenesis in a *Drosophila* model (*Bonnay et al., 2020*). Using analyses of single cells, we reported that neoplastic ovarian cells with minimum mitochondrial fission are found within a (Aldh marked) neoplastic stem cell sub-population that has >10-fold higher self-renewal and proliferation ability compared to the other (Aldh marked) neoplastic stem cell sub-population (*Spurlock et al., 2019*; *Spurlock et al., 2021*). Thus, we proposed, that a repressed mitochondrial fission state may prime neoplastic stem cells toward maximizing their potential of self-renewal and proliferation. Similar priming was demonstrated in a hematopoietic stem cell subpopulation that maintains Drp1 repression (*Liang et al., 2020*).

Here, we investigated the causative role of repression of Drp1-driven mitochondrial fission in mitochondria driven priming of a stem/progenitor-like like state in skin cells that serve as an excellent model for adult stem cells (*Fuchs, 2016*). We established a transformed skin keratinocyte model that sustains abundant self-renewing/proliferating cells with robust repression of Drp1 activity in comparison to the core regulators of mitochondrial fusion. Using single-cell analyses and genetic perturbation strategies we demonstrate for the first time that a 'goldilocks' level of Drp1 repression maintains smaller fused mitochondrial network and a characteristic gene expression profile to support stem/progenitor-like state-dependent neoplastic transformation.

## Results and discussion

### Establishment of a mild carcinogen transformed keratinocyte model that maintains fine-tuned repression of Drp1, smaller fused mitochondria and abundant self-renewing/proliferating cells

Neoplastic transformation can be modeled using chemical carcinogens like 2,3,7,8-Tetrachlorodibenzo-p-dioxin (TCDD) on skin tissue or the keratinocyte HaCaT cell line carrying non-functional p53 (*Boelsma et al., 1999*; *Der Vartanian et al., 2019*; *Hao et al., 2012*; *Jung et al., 2016*; *Ray and Swanson, 2004*; *Schoop et al., 1999*; *Wincent et al., 2012*; *St John et al., 2000*). Given Drp1 repression increases cell proliferation in the absence of active p53 (*Mitra et al., 2012*; *Mitra et al., 2009*), the HaCaT cell model provides the appropriate cellular context for our study involving Drp1. Here, we used both mild and strong dose of TCDD to uncover any mitochondria-based priming of stemness during TCDD driven neoplastic transformation. Exposure to milder (T-1nM) and stronger (T-10nM) doses of TCDD increases cell proliferation of HaCaT cells comparably in a standard transformation protocol (*Figure 1A*). Even in the absence of TCDD, the transformed colonies of TCDD-1nM (TF-1) and TCDD-10nM (TF-10) maintain comparable upregulation of the aryl hydrocarbon receptor AHR (*Figure 1B*), which is the primary effector of TCDD (*Leclerc et al., 2021*; *Mulero-Navarro and Fernandez-Salguero, 2016*). Unlike the parental HaCaT cells (Parental), both transformed cells are able to form subcutaneous xenograft tumors, confirming their transformation status (*Figure 1—figure supplement 1A*). Interestingly, pathological evaluation of H&E stained tumor sections revealed that the TF-1 cells gave rise to malignant tumors harboring less differentiated (primitive) squamous cells with large nuclei (blue) and little visible cytoplasm (pink) (*Figure 1C*, left). Whereas the TF-10 cells formed tumors harboring differentiated stratified squamous epithelium with cells having expansion of cytoplasm (*Figure 1C*, right); the clear gaps represent cryo-sectioning artifacts.

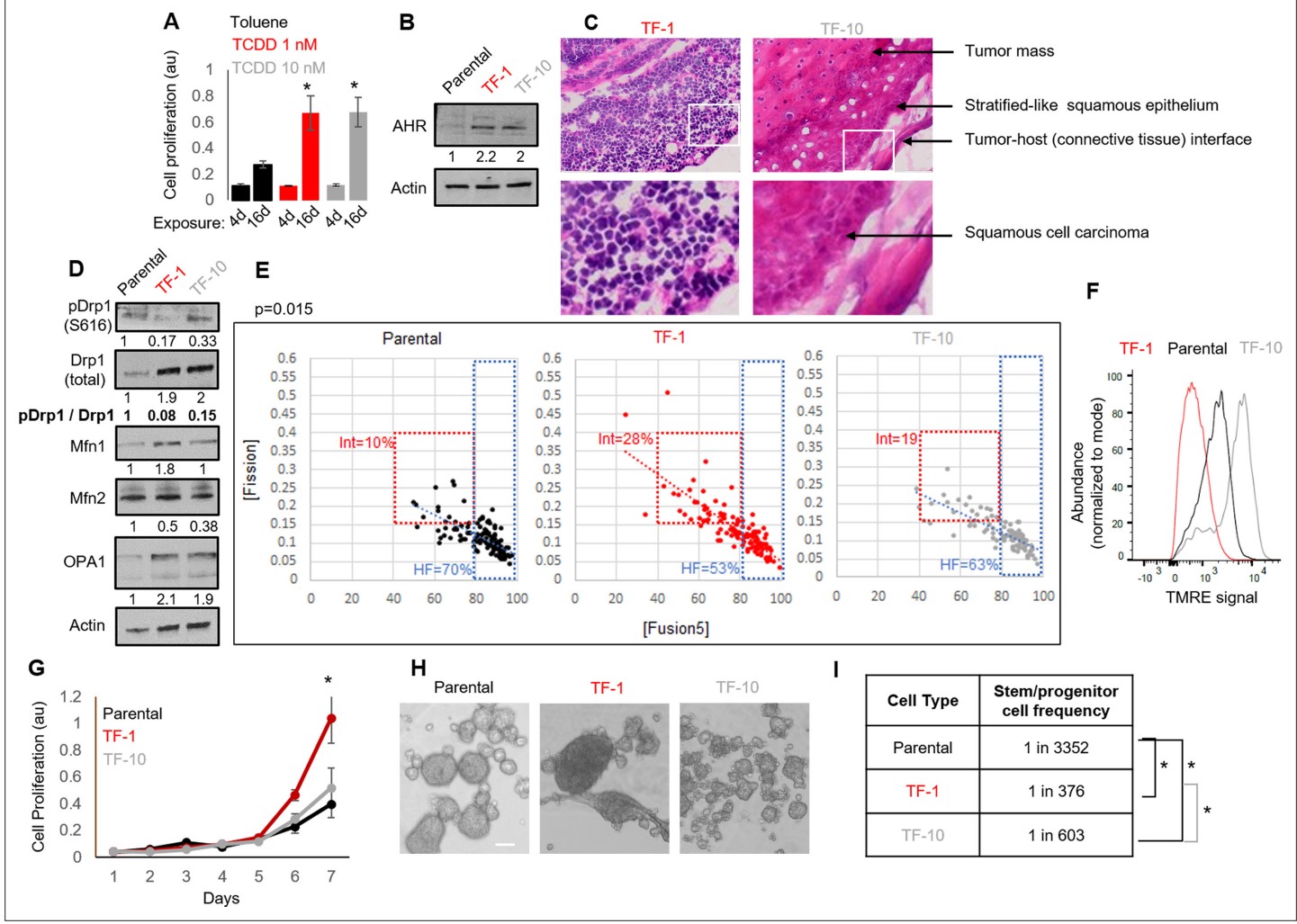

**Figure 1.** Establishment of a mild carcinogen transformed keratinocyte model that maintains abundant self-renewing/proliferating cells and repressed Drp1 activity. (**A**) Cell proliferation assay (quantified by crystal violet staining) after 4 or 16 days exposure to TCDD (1 or 10 nM) or Toluene (10 nM) vehicle control. (**B**) Immunoblot analyses of AHR and actin (loading control) in the absence of TCDD. (**C**) Representative micrographs (40 X) of H&E stained sections of tumor xenografts; zoom of boxed region shown in the bottom panels. (**D**) Immunoblot analyses of mitochondrial fission/fusion proteins with actin as loading control. (**E**) Dot plot of [Fission] and [Fusion5] obtained from confocal micrographs of Mitotracker green stained cells; numbers denote the percentage of cells in the colored boxes with p value from Chi-Square test (N > 90 cells in each group); representative images in **Figure 1—figure supplement 1E**. (**F**) Flowcytometric histogram plots of TMRE intensity of stained cells. (**G**) Quantification of cell proliferation assay (as in A) over 7 days in the absence of TCDD. (**H**) Representative micrograph showing spheroids formed when $10^4$ cells of each group were maintained in low attachment plate in the presence of stem cell medium. (**I**) ELDA based quantification of spheroid forming frequency in each cell population. * signifies p value of < 0.05 in T test (**A,G**) and ELDA (**I**); scale bar depicts 100 µm (**H**); numbers under immunoblots denote quantification of band intensities normalized by that of Actin.

The online version of this article includes the following figure supplement(s) for figure 1:

**Figure supplement 1.** Further characterization of Parantal, TF-1 and TF-10 HaCaT cells.

Notably, among the major proteins regulating mitochondrial fission and fusion (Drp1, Mfn1, Mfn2, Opa1), Drp1 exhibit the most robust difference between the Parental, TF-1 and TF-10 cells (**Figure 1D**, **Figure 1—figure supplement 1C**); other tested mitochondrial markers are lowered in the TF-1 compared to the TF-10 cells (**Figure 1—figure supplement 1B**). In the transformed cells, the activity of the elevated Drp1 protein (pDrp1-S616/ Drp1) is suppressed by ~10 -fold by lowering its cell cycle driven activating phosphorylation (**Taguchi et al., 2007**; **Figure 1D**). Such repression of Drp1-driven mitochondrial fission is expected to allow unopposed mitochondrial fusion towards sustaining elevated mitochondrial matrix continuity. Indeed, the transformed cells maintain increased mitochondrial matrix continuity (**Figure 1—figure supplement 1D**), assessed by live cell pulse chase

assay in cells stably expressing photoconvertible mitochondrial reporter mito-PSmO (*Spurlock et al., 2019*). Unexpectedly, TF-1 cells with twofold less Drp1 activity than TF-10 cells (*Figure 1D*) still maintain lower mitochondrial matrix continuity (*Figure 1—figure supplement 1D*), the cause of which was not apparent from subjective examination of mitochondrial shape (*Figure 1—figure supplement 1E*). Therefore, we performed quantitative assessment of mitochondrial shape that is determined by the opposing mitochondrial fission and fusion processes. We used our previously designed mitochondrial [Fission] and [Fusion5] metrics that quantify the contribution (but not kinetics) of fission/fusion to the steady state mitochondrial shape in live cells, which were validated using Drp1 or mitofusin knockout cells (*Spurlock et al., 2019*; *Spurlock and Mitra, 2021*).Also, bivariate analyses of the inversely related [Fission] and [Fusion5] metrics could reveal specific heterogeneity in mitochondrial shape in (FACS sorted) mitochondria primed ovarian neoplastic stem cell population (*Spurlock et al., 2021*). Here, bivariate analyses of the [Fission] and [Fusion5] metrics revealed that the reduced mitochondrial matrix continuity in TF-1 cells is due to maximum enrichment of a cellular subpopulation with moderately reduced [Fusion5] (40–80), moderately elevated [Fission] (0.15–0.4) (*Figure 1E*, Int), and weakened [Fission] vs [Fusion5] inverse correlation (*Figure 1E*, increased scatter around the regression line). The enrichment of this intermediate fission/fusion subpopulation was accompanied by a proportional reduction of the subpopulation with hyperfused mitochondria ([Fusion5] > 80) (*Spurlock et al., 2019*; *Figure 1E*, HF). Notably, despite maintaining higher mitochondrial matrix continuity than the Parental cells (*Figure 1—figure supplement 1D*), TF-1 population has lowest abundance of cells where the longest mitochondrial element contains >80% of the total mitochondrial length ([Fusion1] > 80) (*Figure 1—figure supplement 1F*). These data together suggest that the enriched subpopulation of TF-1 cells maintains smaller fused mitochondrial elements. These mitochondrial structural properties in the TF-1 cells are associated with overall lowering of mitochondrial transmembrane potential in comparison to Parental or the TF-10 cells (*Figure 1F*). Furthermore, the mitochondrial potential per unit mitochondrial mass in individual cells is regulated over a wider range in TF-1 cells, while it is lowest in TF-10 cells (*Figure 1—figure supplement 1G*) likely due to their elevated mitochondrial mass (*Figure 1—figure supplement 1B*).

The TF-1 cells have significantly higher in vitro cell proliferation rate and form markedly larger spheroids in conditions that support self-renewal and proliferation, in comparison to those formed by Parental or TF-10 cells (*Figure 1G and H*). More importantly, ELDA statistics applied on spheroid formation assay for determination of in vitro stem cell frequency *Hu and Smyth, 2009* demonstrated that both the transformed populations have one order higher abundance of self-renewing/proliferating cells compared to the Parental, while TF-1 has double the abundance than TF-10 population (*Figure 1I*).

The above data confirms the derivation of a transformed TF-1 keratinocyte model characterized by the following properties: (a) fine-tuned repression of Drp1 activity by reduction of S616 phosphorylation of the elevated Drp1 protein level; (b) enrichment of a subpopulation of cells maintaining smaller fused mitochondrial networks; (c) enrichment of self-renewing/proliferating neoplastic cells capable of forming less differentiated primitive tumors. We speculate that twofold lower abundance of self-renewing TF-10 cells is due to their 2-fold elevated Drp1 activity that promotes differentiation in other systems (*Spurlock et al., 2020*), making TF-10 cells more differentiated than TF-1 cells.

## Neoplastic stem cell enriched TF-1 keratinocyte population, with fine-tuned Drp1 repression, maintains an expanded sub-population of slow-cycling cells expressing elevated stem/progenitor cell markers

We performed scRNA-seq to define the neoplastic stem cells of our newly derived Drp1-repressed-TF-1 population (*Luecken and Theis, 2019*). Cell clustering in a UMAP plot shows six clusters within Parental, TF-1 and TF-10 populations (*Figure 2A*, Clusters 0–5). While Cluster 4 is reduced in both transformed populations in comparison to the Parental, the TF-1 population exhibits a marked expansion of Cluster 3 and reduction of Cluster 5 (*Figure 2A*, *Figure 2—figure supplement 1A*, left panel). These results also hold true with the lowest cluster resolution (*Figure 2—figure supplement 1B*, arrows). We identified the top-most candidate genes to mark each cluster (color coded arrows in *Figure 2—figure supplement 1C*, *Figure 2—source data 1*). Indeed, the neoplastic stem/progenitor cell enriched TF-1 population shows >3 fold upregulation of the epidermal stem cell marker Keratin(*KRT*)15 (*Giroux et al., 2017*; *Purba et al., 2014*) and of the neoplastic stem cell markers for

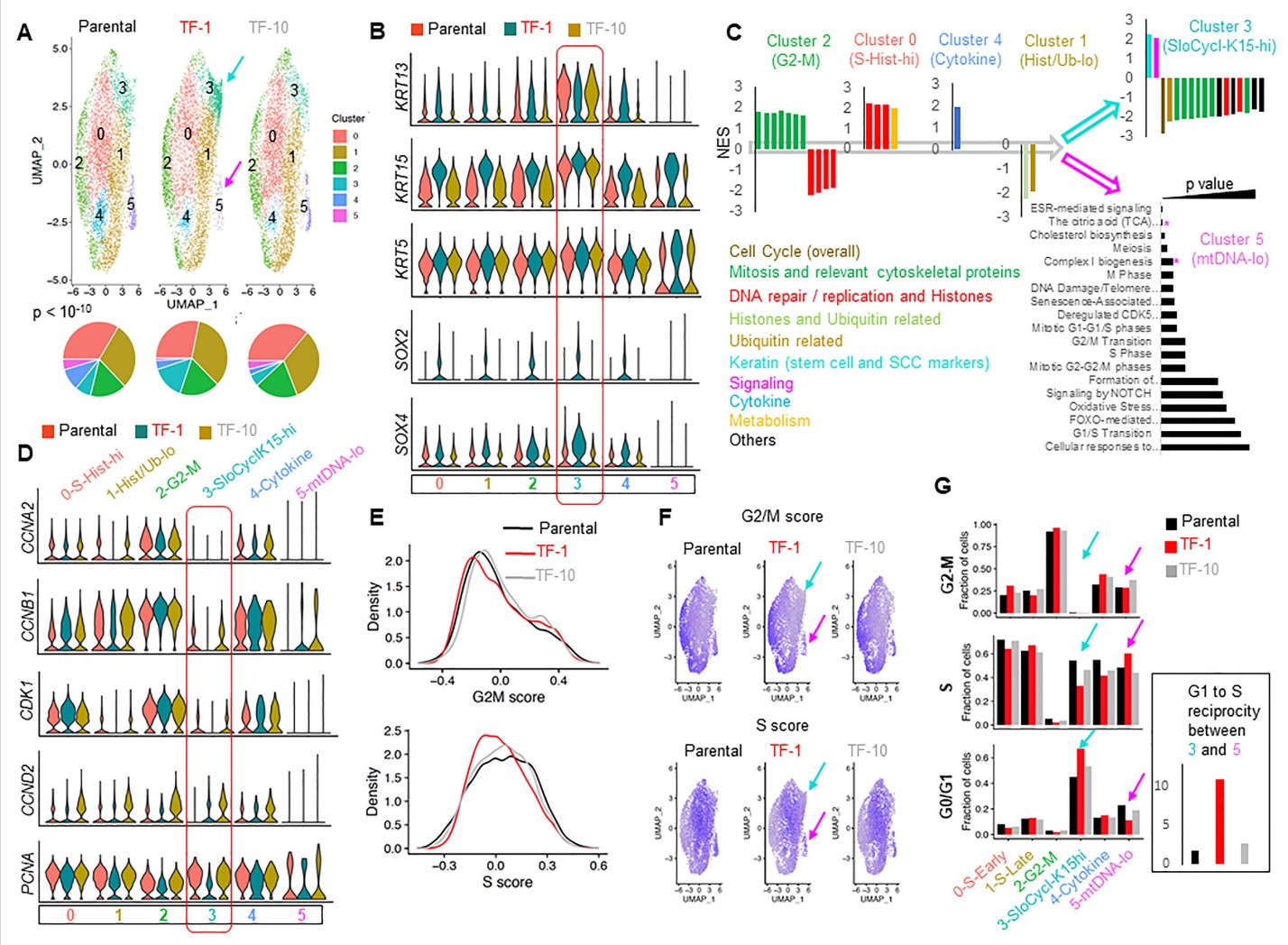

**Figure 2.** Drp1-repressed-TF-1 transformed keratinocyte population maintains an expanded sub-population of slow-cycling cells expressing elevated stem/progenitor cell markers. (**A**) UMAP plot of scRNA-seq derived clusters (0–5) in individual cell population (top); pie chart of the percentage distribution of the clusters (bottom, p value from Chi-Square test); color coded arrows mark the clusters with unique reciprocal abundance in TF-1 cells. (**B**) Violin plots depicting expression of specific stem/progenitor cell markers in the clusters identified in (**A**); box outlines the *KRT15*-hi Cluster 3. (**C**) Normalized Enrichment Scores (NES) of functional pathways identified by GSEA of marker genes of Clusters 0–4 (***Figure 2—source data 2***) and by overrepresentation analyses for Cluster 5 (* represents categories including mtDNA genes); functional categories defined by majority of the leading-edge genes are in parentheses, while clusters are arranged based on trajectory analyses (open arrows) (see ***Figure 2—figure supplement 1F***). (**D**) Violin plots depicting expression of key cell cycle regulators in the clusters identified in (**A**); the functional category of genes identified in (**C**) is included in the cluster identity, while box outlines the *KRT15*-hi Cluster. (**E**) Distribution of G2-M and S scores obtained from gene expression analyses of cell cycle predictive genes. (**F**) Feature plot of G2-M and S scores obtained from (**E**); color coded arrows as in (**A**). (**G**) Bar plot showing fraction of cells in G2-M, S and G0/G1, computed from (**E**) in the functionally categorized clusters; inset shows G1 to S reciprocity of the clusters (color-coded arrows) with unique reciprocal abundance in TF-1 cells in (**A**).

The online version of this article includes the following source data and figure supplement(s) for figure 2:

**Source data 1.** Cluster markers of Parental, TF-1 and TF-10 HaCaT cells.

**Source data 2.** GSEA of cluster markers of Parental, TF-1 and TF-10 HaCaT cells.

**Figure supplement 1.** Further sc-RNAseq analyses of Parental, TF-1 and TF-10 HaCaT cells.

skin carcinomas, *SOX2* and *SOX4* (***Boumahdi et al., 2014***; ***Foronda et al., 2014***) (***Figure 2B*** (box)). Since the markers increase across all TF-1 clusters, with maximum levels in Cluster 3, we conclude that the TF-1 population exhibits upregulation of the stem/progenitor markers independent of their cluster distribution. On the other hand, the other stem/progenitor cell markers, like *KRT5*, *KRT14*, *KRT19* (***Gonzales and Fuchs, 2017***, ***Karantza, 2011***) are also enriched by ~2 -fold in Cluster 3, but

comparably in all three cell populations, as exemplified by *KRT5* (*Figure 2B*, box, *Figure 2—source data 1*). Notably, among the three -cell populations, TF-1 harbors markedly fewer cells with elevation of the differentiating cell marker *KRT13* in Cluster 3 (*Karantza, 2011*). Cluster 5, reduced in TF-1 population, characterized by reduction of overall gene expression, is marked by 1.5 to >2 -fold reduced expression of 12 of the 13 mtDNA genes in all the three-cell populations (*Figure 2—figure supplement 1C, D* (box)). Cluster 4, reduced in both transformed populations, is marked by >8 -fold upregulation of *OASL*, a gene involved in the interferon pathway, in all the three cell populations (*Figure 2—figure supplement 1E*).

Since flow cytometry on PI stained cells revealed modest differences in cell cycle distribution between the Parental and transformed populations (*Figure 2—figure supplement 1A*, right panel), we investigated if the scRNA-seq-derived clusters represent distinct cell cycle phases. We performed gene set enrichment analyses (GSEA) of the overall cluster marker profiles (*Subramanian et al., 2005*); some statistically significant REACTOME pathways (q value < 0.01) were renamed based on their leading-edge genes (*Figure 2—source data 2*). GSEA showed upregulation of genes in 'Mitosis and Relevant Cytoskeletal Proteins' in Cluster 2 and that of genes in 'DNA repair/replication and Histones' in Cluster 0 (*Figure 2C*); Cluster 1 has downregulation of "Histones and Ubiquitin related' genes. Indeed, the mitotic Cyclin B (*CCNB1*) and its partner kinase *CDK1*, as well as Cyclin A (*CCNA2*) are markedly higher in Cluster 2-G2-M, while *PCNA*, which peaks early in S phase (*Maga and Hubscher, 2003*; *Zerjatke et al., 2017*) is highest in Cluster 0-S-Histone-hi in all three-cell populations (*Figure 2D*); the S phase cyclins E1/E2 and their partner *CDK2* were not detected in our scRNA-seq data set. Surprisingly, *KRT15* marked Cluster 3 has dramatic downregulation of genes in 'Overall Cell Cycle' and in cell cycle phase categories described above (*Figure 2C*), suggesting their cell cycling is slow. Cluster 3-SloCycl-K15-hi maintains extreme low levels of major cyclins in all three-cell populations, most prominently in TF-1 (*Figure 2D*). PCNA, which is dramatically reduced in quiescent cells (*Maga and Hubscher, 2003*; *Zerjatke et al., 2017*), is markedly reduced in the SloCycl-K15-hi cluster (and others) only in the TF-1 population (*Figure 2D*). SloCycl-K15-hi has upregulation of genes involved in 'Keratin' and 'Signaling' pathways (*Figure 2C*). Notably, the Notch pathway target, *HES1*, which actively maintains quiescence (*Moriyama et al., 2008*; *Sang et al., 2008*), is upregulated across the clusters in the TF-1 population (*Figure 2—figure supplement 1E*). These data strongly suggest that the enrichment of the SloCycl-K15-hi cluster in the TF-1 population happens due to active maintenance of quiescence, rendering them slow cycling. Cluster four does not relate to cell cycle and shows upregulation of genes related to 'Cytokine' (interferons) pathway (*Figure 2C*), many of which are suppressed by AHR activation (*Di Meglio et al., 2014*; *Figure 2—source data 1*). This finding is consistent with our findings in *Figures 1B and 2A*. The downregulated mtDNA genes in Cluster 5 were identified as 'TCA cycle' and 'Complex one biogenesis' in our overrepresentation analyses (* in *Figure 2C*), but not detected in GSEA. Towards understanding the entry into the slow cycling state of the K15hi-Cluster 3, we performed trajectory analyses using Slingshot algorithm that identifies clusters related to each other based on their gene expression (*Street et al., 2018*). The output trajectory was the following: 'G2-M' to 'S-Histones-hi' to 'Cytokine' to 'Histones/Ub-lo' bifurcating into 'SloCycl-K15-hi' or 'mtDNA-lo' (*Figure 2C*, open arrows, *Figure 2—figure supplement 1F*). This implies that the cells in the TF-1 population may preferentially reside in the SloCycl-K15-hi cluster and not in the mtDNA-lo cluster, thus expanding the former and reducing the latter (*Figure 2A*).

To confirm the cell cycle pathway findings from GSEA, we quantified cell cycle scores and distribution in each cluster using a Seurat algorithm that employs several validated predictive cell cycle genes (*Tirosh et al., 2016*). This showed that the TF-1 population has lower S score than the Parental and the TF-10 populations, while G2-M scores were comparable (*Figure 2E*). Indeed, Cluster 2 is abundant in G2-M cells, Cluster 0 (and 1) in S and *KRT15* marked Cluster 3 is in G0/G1 (expected for quiescent/slow-cycling cells) (*Figure 2F and G*). This data taken with the lower levels of Histone transcripts in Cluster 1 than in Cluster 0 (* in *Figure 2—figure supplement 1C*) suggest that Cluster 0 may be in early S phase while Cluster 1 in late S phase. We noted that the expanded 3-SloCycl-K15-hi cluster in the TF-1 population has significantly more cells in G0/G1 and less cells in S than the other two populations (*Figure 2F and G*, arrows), which is opposite to the reduced 5-mtDNA-lo cluster in the TF-1 population (*Figure 2F and G*, arrows). This G1 to S reciprocity between Cluster 3 and 5 (quantified from the distributions as in Materials and methods) is >5 fold in the TF-1 population with respect to the Parental and TF-10 population (*Figure 2G*, inset). This suggests that altered cell cycle regulation in

the TF-1 population may allow them to preferentially reside in the 3-SloCycl-K15-hi cluster and not in the 5-mtDNA-lo cluster. Such a regulation may involve attenuation of CDK1-driven Drp1(S616) phosphorylation (*Figure 1D*) due to marked reduction of *CCNB1/CDK1* particularly in the TF-1 population in late S phase (*Figure 2D*), where decision for expansion of the slow cycling *KRT15*-hi cluster is likely made (*Figure 2C*, *Figure 2—figure supplement 1F*).

Thus, detailed scRNA-seq characterization reveals that our newly derived Drp1-repressed-TF-1 keratinocyte population, enriched with neoplastic stem/progenitor cells, maintains an expanded sub-population of slow cycling cells expressing high levels of stem/progenitor markers (lineage specific: *KRT15*; general: *SOX2* and *SOX4*), likely due to altered cell cycle regulation. Our data is consistent with the *KRT15*-hi epidermal stem cells being primarily slow cycling (*Giroux et al., 2017*; *Morris et al., 2004*; *Purba et al., 2014*). Moreover, the higher in vitro cell proliferation rate of the Drp1-repressed-TF-1 keratinocytes (*Figure 1G*) could result from the higher clonogenic capacity of the *KRT15*-hi cells (*Morris et al., 2004*; *Seykora and Cotsarelis, 2011*).

## Fine-tuning Drp1 by reducing its S616 phosphorylation sustains a [Sox2$^{hi}$-Krt15$^{hi}$] neoplastic stem/progenitor cell state in TF-1 keratinocytes with elevated cyclin E levels

Repression of Drp1-driven mitochondrial fission elevates stem/progenitor cell markers in multiple cell types (*Iwata et al., 2020*; *Parker et al., 2015*). The TF-1 keratinocytes, characterized by low pDrp1-S616 levels and high enrichment of neoplastic stem/progenitor cells (*Figure 1D, H, I*), exhibit marked elevation in Sox2 protein levels (but no other embryonic stem cell markers) in comparison to the Parental and TF-10 keratinocytes (*Figure 3A*, *Figure 3—figure supplement 1A*); this corroborates the scRNA-seq data (*Figure 2B*). Krt15 protein levels are elevated in both transformed cells in comparison to the Parental (*Figure 3A*), unlike the cluster-specific differences in transcript levels (*Figure 2B*). Immunostaining-based co-expression analyses of Sox2 and Krt15 revealed that ~26 % of the TF-1 population exhibits elevation of both Krt15 and Sox2, compared to ~6 % in Parental (*Figure 3B*). Similar analyses showed that the high levels of Sox2 is sustained in ~26 % of the TF-1 population that has lowest pDrp1(S616) levels, compared to ~6 % in Parental (*Figure 3C*). Thus, our data suggests about a quarter of the TF-1 population maintains low pDrp1, high Sox2 and high Krt15.

Cell cycle status is a crucial determinant of stemness (*Orford and Scadden, 2008*; *Otsuki and Brand, 2018*; *Pauklin and Vallier, 2013*). Sox2 expression is under the influence of various G1 cyclins (*Liu et al., 2019*), including Cyclin E whose degradation is regulated by mitochondria (*Mandal et al., 2010*; *Parker et al., 2015*).Particularly, Drp1 repression elevates Cyclin E levels (*Spurlock et al., 2020*). Consistently, among the major cyclins, Cyclin E1 levels are particularly higher in the Sox2 enriched Drp1 repressed TF-1 cells (*Figure 3D*); Cyclin D is particularly higher in TF-10 cells when compared to the Parental (corroborating scRNA-seq data, *Figure 2D*). Moreover, Cyclin E1 is preferentially accumulated in the nucleus in TF-1 cells, which is required to impact gene expression of Sox2 (and others) (*Figure 3E and F*). The level of CDK2-driven pCyclinE(T-62) reflects the level of CDK2 bound Cyclin E that is susceptible to degradation (*Siu et al., 2012*). Immunostaining based co-expression analyses showed that elevated Sox2 is sustained in ~38 % of the TF1 population that sustain lower pCyclinE(T-62) as opposed to ~2 % in the Parental (*Figure 3G*). Moreover, a cycloheximide chase assay, for assessing protein degradation rate, revealed no Cyclin E degradation in TF-1 cells within the 2 -hr assay time frame in comparison to marked Cyclin E degradation in Parental; both populations maintain Tom20 (mitochondrial marker) (*Figure 3—figure supplement 1B*). Thus, our data is consistent with elevated Sox2 in TF-1 cells being sustained by elevated Cyclin E levels achieved by attenuation of its degradation kinetics, which can be potentially impacted by Drp1 repression (*Parker et al., 2015*).

Next, we investigated if repression of Drp1-S616 phosphorylation maintains Sox2 expression since highest levels of Sox2 is expressed in the cells with reduced pDrp1-S616 (*Figure 3C*). To this end, we stably overexpressed the phospho-deficient Drp1-S616A mutant and the Drp1-WT control in the Parental and TF-1 cells (*Serasinghe et al., 2015*). We confirmed in the Parental cells, Drp1-S616A overexpression maintains markedly reduced pDrp1-S616 levels (*Figure 3H*) and has attenuated ability of elevating mitochondrial [Fission] and lowering [Fusion5/1] in comparison to the Drp1-WT overexpression (*Figure 3—figure supplement 1C*). On the other hand in the TF-1 cells with already reduced basal pDrp1-S616 levels, Drp1-S616A overexpression modestly reduces pDrp1-S616 levels

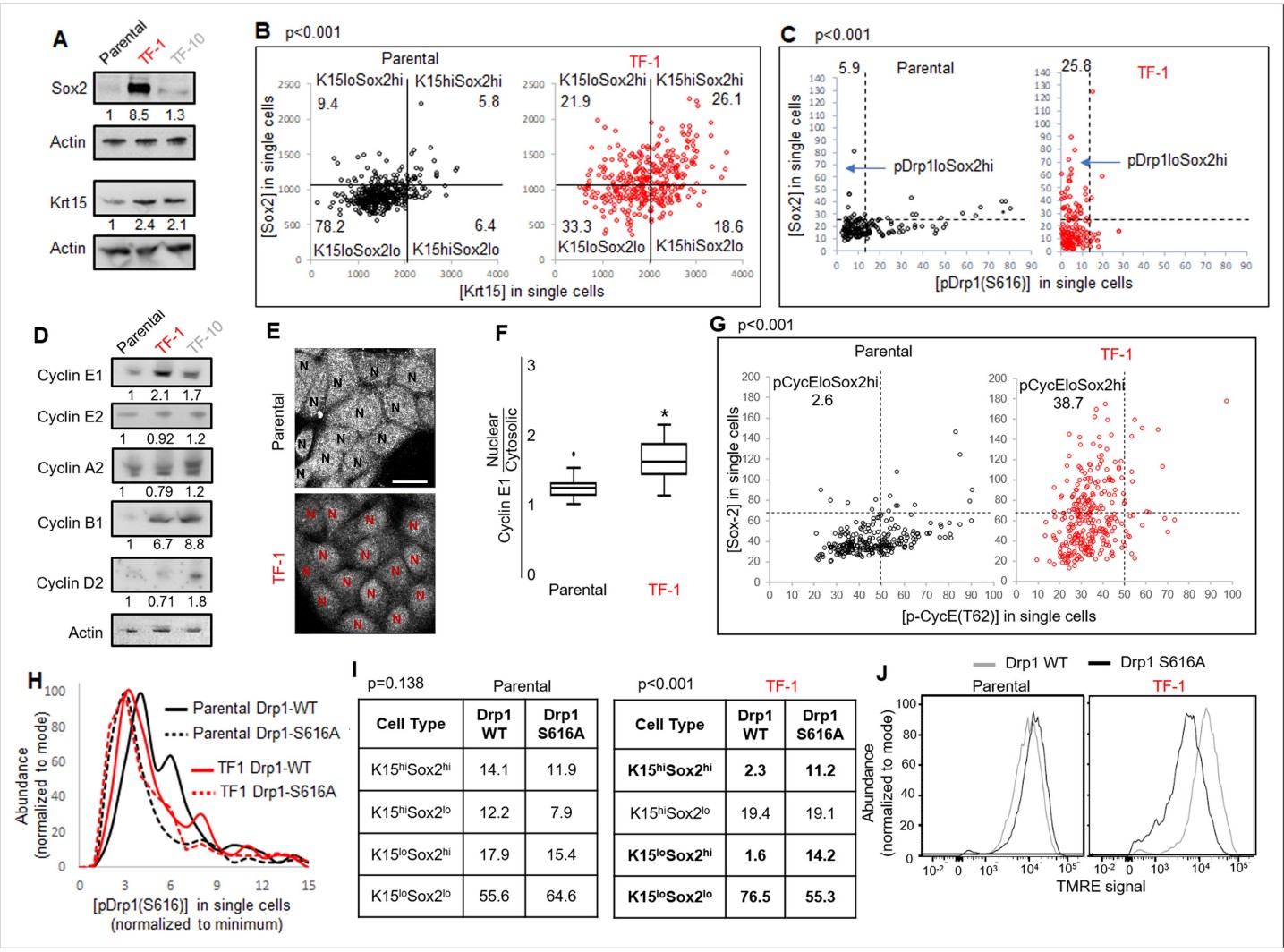

**Figure 3.** Fine-tuning Drp1 by reducing its S616 phosphorylation sustains a [Sox2$^{hi}$-Krt15$^{hi}$] neoplastic stem/progenitor cell state in TF-1 keratinocytes with elevated Cyclin E levels. (**A**) Immunoblot analyses of stem/progenitor cell markers with actin as loading control. (**B**) Dot plot of [Sox2] vs [Krt15] levels obtained from confocal micrographs of co-immunostained cells demarcating the four populations; N > 300 cells in each group. (**C**) Dot plot of [Sox2] and [pDrp1(S616)] levels obtained from confocal micrographs of co-immunostained cells demarcating the population of interest (arrow); N > 200 cells in each group. (**D**) Immunoblot analyses of major cyclins with actin as loading control. (**E**) Representative confocal micrographs of Cyclin E1 immunostaining; N depict the nucleus in each cell identified by DNA stain (not shown). (**F**) Box plot showing median and distribution of nuclear to cytosolic ratio of Cyclin E1 levels quantified from experiment described in (**E**); N > 180 cells in each group. (**G**) Dot plot of [Sox2] and [CyclinE1(pT62)] levels obtained from confocal micrographs of co-immunostained cells; N > 120 cells in each group. (**H**) Histogram plots obtained from quantification of pDrp1-S616 signal from confocal micrographs of immunostained Drp1 modified cells. (**I**) Table denoting abundance (%) of each population identified as in (**B**) in the Drp1 modified cells (see **Figure 3—figure supplement 1D**). (**J**) Flowcytometric histogram plots of TMRE staining intensity of Drp1 modified cells; cell numbers under immunoblots denote quantification of band intensities normalized to that of actin; dotted lines in dot plots represent signal threshold marking the quadrants (described in Materials and methods) with the numbers signifying the abundance (%) of each population and the p value of Chi-Square test mentioned; * signifies p value of < 0.05 in KW test (**F**).

The online version of this article includes the following figure supplement(s) for figure 3:

**Figure supplement 1.** Further analyses of stem/progenitor cell markers, Cyclin E and mitochondrial properties in Parental and TF-1 HaCaT cells with or without genetic modification of Drp1.

(**Figure 3H**) and prevents the paradoxical increase in [Fusion5/1] observed with Drp1-WT overexpression while both do not impact [Fission] (**Figure 3—figure supplement 1C**). This paradox can be explained if specific Drp1 regulators unnaturally sequester Drp1 to allow unopposed mitochondrial fusion (**Palmer et al., 2011**). Nonetheless, this data suggests that the characteristic reduced [Fusion5/1] in the TF-1 cells (**Figure 1E**, **Figure 1—figure supplement 1F**) is maintained by preventing

S616 phosphorylation of their elevated Drp1 protein levels. More importantly, the Drp1-S616A expressing TF1 cells maintains ~5 fold higher abundance of Sox2$^{hi}$ subpopulations (irrespective of their Krt15 levels) than Drp1-WT expressing TF1 cells that maintain higher abundance of Krt15$^{lo}$-Sox2$^{lo}$ subpopulation (*Figure 3I*, *Figure 3—figure supplement 1D*). Also, Drp1-S616A expressing TF-1 cells maintain markedly lower mitochondrial membrane potential in comparison to the Drp1-WT expressing cells (*Figure 3J*), suggesting lower mitochondrial potential in the TF-1 cells (*Figure 1F*) is maintained by lowering pDrp1-S616 levels. Notably, such impact of Drp1-S616A expression on Sox2 or mitochondrial potential is not observed in the Parental cells (*Figure 3I and J*), which maintain higher pDrp1-S616 levels and mitochondrial membrane potential (*Figure 1D and F*). Thus, repression of pDrp1-S616 results in altered mitochondrial potential and higher abundance of the Sox2$^{hi}$-Krt15$^{hi}$ sub-population in the TF-1 but not in the Parental population.

Therefore, our data demonstrates that the neoplastic stem/progenitor cell enriched TF-1 population, with elevated nuclear Cyclin E, maintains a [Sox2$^{hi}$-Krt15$^{hi}$] sub-population that is sustained by fine-tuning Drp1 by reducing its S-616 phosphorylation. Cell-cycle-dependent reduction of Cyclin B1/CDK1 levels in the TF-1 cells (*Figure 2D*) likely reduces their Drp1-S616 phosphorylation (*Taguchi et al., 2007*), which in turn maintains less fused mitochondria and modulates mitochondrial membrane potential to elevate Sox2 levels. It remains to be seen how regulation of pDrp1-S616 impacts Cyclin E toward maintaining elevated Sox2, although Drp1 regulation appears to enhance Cyclin E levels particularly in mitotic TF-1 (*Figure 3—figure supplement 1E*). Our data also reveal differences in the impact of Drp1 regulation between the non-transformed Parental and its transformed TF-1 counterpart, which warrants further detailed investigation.

## Fine-tuned Drp1 repression by reducing its protein levels sustains a [CyclinE$^{hi}$-Sox2$^{hi}$-Krt15$^{hi}$] state and accelerates carcinogen driven neoplastic transformation, which is suppressed by more complete Drp1 repression

Given Drp1-S616A mutant does not elevate Sox2 or Krt15 in the non-transformed Parental cells (*Figure 3I*), we hypothesized that reduction of their Drp1 protein levels allows expression of the stem/progenitor cell markers as noted in Drp1 knockout cells of other cell lineages (*Iwata et al., 2020*; *Parker et al., 2015*). Therefore, we knocked down (kd) Drp1 expression in the Parental cells with two validated shRNAs against human *DNM1L* gene encoding Drp1 (*Parker et al., 2015*; *Tanwar et al., 2016*), one weaker (W) and the other ~10-fold stronger (S) almost mimicking Drp1 genetic ablation (*Figure 4A*). Surprisingly, while transient weaker Drp1 knockdown markedly elevates Krt15 levels, the transient stronger Drp1 knockdown does so only modestly (*Figure 4A*). On the other hand, both weak and strong transient Drp1 knockdown cause marked elevation of Sox2 (*Figure 4A*) and dramatic nuclear accumulation of Cyclin E (*Figure 4B and C*). The majority of the cytosolic Cyclin E particularly in the Drp1 knockdown cells represent mitochondrial Cyclin E (not shown) (*Parker et al., 2015*), which remains to be investigated. Therefore, unlike close to complete Drp1 knockdown, fine-tuned knockdown of Drp1 protein levels with the weaker shRNA maintains a CyclinE$^{hi}$-Sox2$^{hi}$-Krt15$^{hi}$ state characteristic of the naturally Drp1 repressed TF-1 population enriched in neoplastic stem/progenitor cells.

Given, Sox2 and Cyclin E are important for neoplastic transformation in various cancer types (*Hwang and Clurman, 2005*; *Schaefer and Lengerke, 2020*; *Teixeira and Reed, 2017*), and Krt15-hi cells give rise to skin carcinoma (*Li et al., 2013*; *Seykora and Cotsarelis, 2011*), we tested if repression of Drp1 can modulate the process of carcinogen driven neoplastic transformation of keratinocytes. Thus, we exposed control and Drp1 knockdown Parental keratinocytes to 1 and 10 nM TCDD following standard TCDD driven transformation protocol (similar to *Figure 1A*). Interestingly, the weaker Drp1 knockdown, which maintains remnant Drp1 protein and a CyclinE$^{hi}$-Sox2$^{hi}$-Krt15$^{hi}$ state, was transformed by 10 nM TCDD even earlier than the control (*Figure 4D*); 1 nM TCDD did not cause any transformation at this time point. This happened despite their lower cell proliferation rate than the control in the earlier time point. On the other hand, the stronger Drp1 knockdown, which also maintained slower cell proliferation rate and a Krt15$^{lo}$-Sox2$^{hi}$ status with elevated nuclear Cyclin E, is not transformed in the same time window (*Figure 4D*). Furthermore, in case of lowered knockdown efficacy, both the weak (W-L) and strong (S-L) Drp1 knockdown (*Figure 4E*) increase abundance of self-renewing cells (*Figure 4F*) and initiate TCCD-driven neoplastic transformation at the earlier time point when control knockdown cells do not exhibit any neoplastic transformation (*Figure 4G*); this

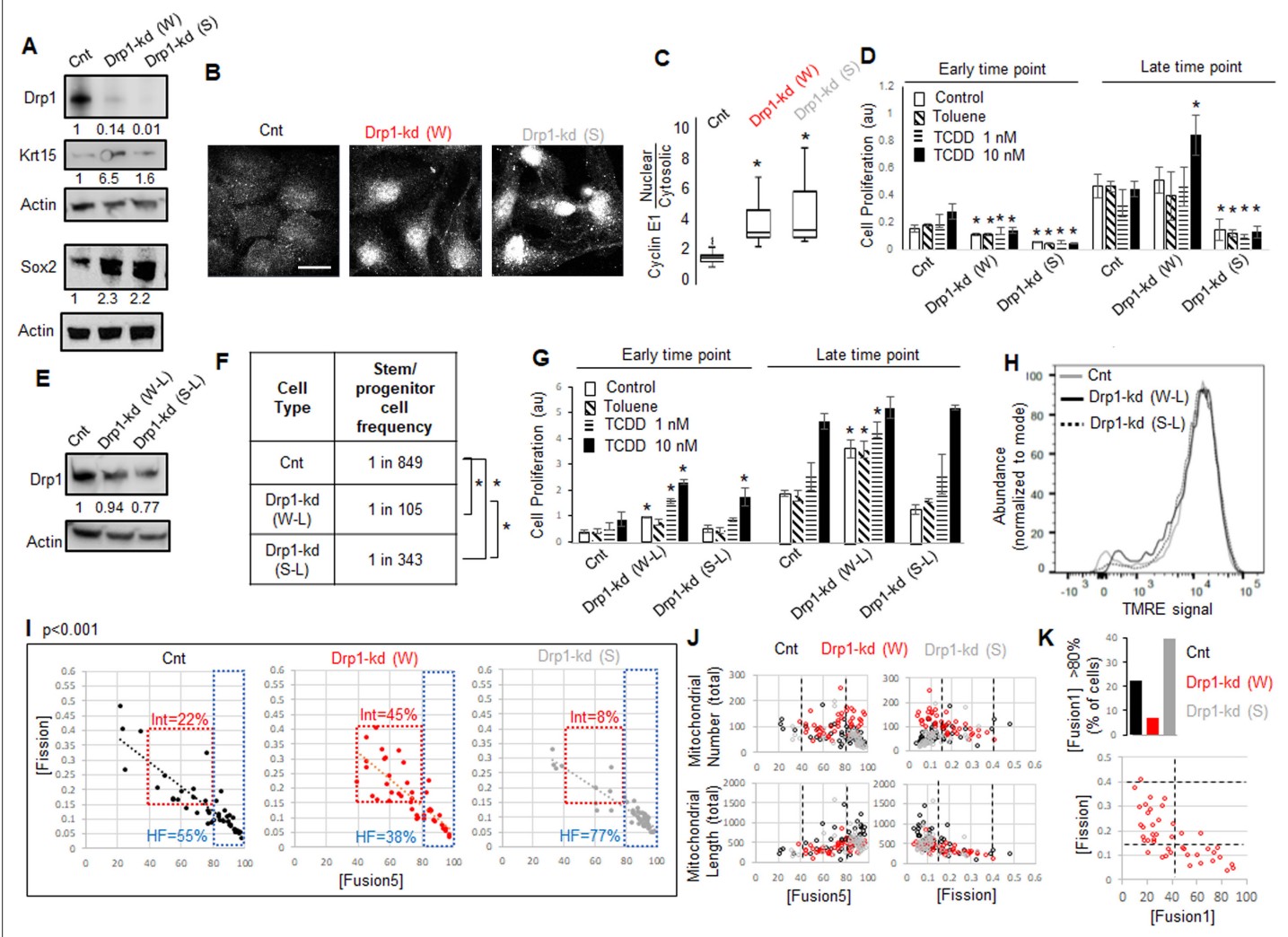

**Figure 4.** As opposed to more complete Drp1 repression, fine-tuned Drp1 repression sustains a [CyclinE<sup>hi</sup>-Sox2<sup>hi</sup>-Krt15<sup>hi</sup>] state, accelerates carcinogen-driven keratinocyte transformation, and maintains smaller fused mitochondria. (**A**) Immunoblot analyses of control (Cnt), weaker (**W**) or stronger (**S**) Drp1 knockdown (kd) Parental cells, with actin as loading control. (**B**) Representative images of Cyclin E1 immunostaining in control or Drp1 knockdown Parental cells. (**C**) Box plot showing median and distribution of nuclear to cytosolic ratio of Cyclin E1 levels quantified from experiment described in (**B**). (**D**) Cell proliferation assay of control or Drp1 knockdown Parental cells after exposure to TCDD (1 nM or 10 nM) at an early (10d) and a late time point (17d) in the neoplastic transformation protocol; 'Toluene' is the vehicle control for TCDD and 'Control' is with no chemical. (**E**) Immunoblot analyses of Parental cells with control or reduced level of Drp1 knockdown with weak (**W–L**) or strong (**S–L**) *DNM1L* shRNAs, with actin as loading control. (**F**) ELDA based quantification of spheroid forming frequency of cells described in (**E**). (**G**) TCDD induced cell proliferation assay (as in D) of cells described in (**E**). (**H**) Flowcytometric histogram plots of TMRE staining of cells described in (**E**). (**I**) Dot plot of [Fission] and [Fusion5] obtained from confocal micrographs of Mitotracker green stained live control or Drp1 knockdown Parental cells described in (**A**); numbers denote the percentage of cells in colored boxes with p value from Chi-Square test (N > 40 cells in each group); representative images in *Figure 3—figure supplement 1D*. (**J**) Dot plot of [Fission] or [Fusion5] of single cells (from I) with their total mitochondrial number and total mitochondrial length; lines represent the ranges of [Fission] and [Fusion5] to demarcate cell subpopulations in boxes in (**I**). (**K**) Bar graph signifying percentage of cells from (**I**) with >80 [Fusion1] metric (top); dot plot of [Fission] vs [Fusion1] of cells from (**I**) (bottom); cell numbers under immunoblots denote quantification of band intensities normalized to that of actin; * signifies p value of < 0.05 in KW test (**C**), T test (**D,G**) and ELDA (**F**); scale bar depicts 20 μm (**B**).

The online version of this article includes the following figure supplement(s) for figure 4:

**Figure supplement 1.** Further analyses of transformation efficacy and mitochondrial shape of Drp1 knockdown HaCaT cells.

difference between control and the low efficacy Drp1 knockdown does not sustain in the later time point. Notably, the accelerated carcinogen-driven neoplastic transformation caused by Drp1 knockdown does not involve modulation of mitochondrial potential (*Figure 4H*).

Therefore, we conclude that fine-tuned reduction of Drp1 protein levels in the Parental cells primes the 'stem/progenitor-like [CyclinE^{hi}-Sox2^{hi}-Krt15^{hi}] state' toward accelerating neoplastic transformation, which is suppressed by more complete Drp1 reduction. We speculate that the inability to sustain this primed stem-like state may prevent neoplastic transformation in Drp1 knockout MEFs (*Serasinghe et al., 2015*). Plotting the accelerated transformation efficacy with remnant Drp1 levels after knockdown predicts ~50 % reduction of Drp1 protein may maximally accelerate transformation within the experimental range (*Figure 4—figure supplement 1A*) (remnant Drp1 levels may remain overestimated due to the reduction of the Actin control with Drp1 knockdown, *Figure 4A and E*).

## Fine-tuned Drp1 repression maintains smaller fused mitochondria and transcriptomic profile characteristic of the neoplastic stem/progentior cell enriched TF-1 population, which is suppressed by more complete Drp1 repression

Toward understanding how the 'stem/progenitor-like state' is primed by fine-tuned (incomplete) Drp1 repression, we performed detailed comparison of mitochondrial shape (as in *Figure 1E*) and over all gene expression (as in *Figure 2*) between the Parental keratinocytes expressing control, weak or strong *DNM1L* shRNAs. Bivariate analyses of the inversely related mitochondrial [Fission] and [Fusion5] metrics confirmed that the stronger (more complete) Drp1 knockdown maintains ~77% cells in the uniformly hyperfused state while the control knockdown population maintains ~55% cells in this state (*Figure 4I*, HF, *Figure 4—figure supplement 1D*). Interestingly, similar to the neoplastic stem/ progenitor cells enriched Drp1 repressed TF-1 cells (*Figure 1E*), the ~10 -fold weaker Drp1 knockdown maintains only ~38% cells in the uniformly hyperfused state, while enriching a sub-population of cells (~45%) with moderately increased [Fission] and moderately decreased [Fusion5] (*Figure 4I*, Int). Subjective examination revealed that cells in the enriched sub-population (Int, Fission: 0.15–0.4 and Fusion5: 40–80) harbor some mitochondrial fragments in the midst of tubular mitochondrial network (*Figure 4—figure supplement 1B*). We also found that the cells in this sub-population have lower mitochondrial number and total mitochondrial length (*Figure 4J*), signifying decrease in mitochondrial content. Also, similar to the TF-1 cells (*Figure 1—figure supplement 1F*), ~ 7 % of cells in the weaker Drp1 knockdown have the longest mitochondrial element containing >80% of the total mitochondrial length ([Fusion1] > 80), compared to ~40 % in the stronger Drp1 knockdown (*Figure 4K*, top, *Figure 1—figure supplement 1C,D*). This indicates that the increase in mitochondrial number is due to a greater number of smaller networks of fused mitochondria, consistent with findings in TF-1 cells. Moreover, bivariate [Fission] vs [Fusion1] plot revealed that the cells with fission range of 0.15–0.4, enriched in the weaker Drp1 knockdown and TF-1 cells, have only up to 40 % of the mitochondrial lengths in fused networks (*Figure 4K*, bottom). Given Drp1-WT overexpression in the Parental cells expectedly increases [Fission] and reduces [Fusion] (*Figure 3—figure supplement 1C*), our data suggest that the remnant Drp1-driven mitochondrial fission after fine-tuned Drp1 repression prevents mitochondrial hyperfusion to maintain smaller fused mitochondria (with reduced mitochondrial content), towards sustaining a [CyclinE^{hi}-Sox2^{hi}-Krt15^{hi}] state. However, Drp1 overexpression by itself does not support Sox2hi/Krt15hi state in Parental cells (*Figure 3I*) as does weaker repression of Drp1 (*Figure 4A*), suggesting Drp1 needs to be repressed to support expression of stem cell markers and then further activated. Indeed, Aldh marked neoplastic ovarian stem cells with repressed [Fission] converts to a state of elevated mitochondrial [Fission] when self-renewal and proliferation is activated (*Spurlock et al., 2019*), which may be triggered to support mitophagy (*Twig and Shirihai, 2011*).

To probe the contribution of gene expression of any particular cell population in sustaining a stem/ progenitor-like state after fine-tuned Drp1 repression, we performed scRNA-seq of the control and Drp1 knockdown Parental populations. Analyses of the overall gene expression (within the limits of gene coverage) confirmed that the weaker Drp1 knockdown, which maintains higher transcript levels of *DNM1L* (*Figure 5—figure supplement 1A*) and ~10 -fold higher protein levels (*Figure 4A*), modulates a smaller gene set to a smaller degree (*Figure 5—figure supplement 1B*). However, graded Drp1 knockdown has both 'similar' (graded upregulation or downregulation) and 'opposite' effects (upregulation in one and downregulation in the other) (*Figure 5—figure supplement 1C*).

The opposite effect is dominated by *KRT15*, *KRT13*, *KRT4*, *KRT5* transcripts being upregulated in the weaker Drp1 knockdown and downregulated in the stronger Drp1 knockdown (color coded * in *Figure 5A*, *Figure 5—figure supplement 1C*, *Figure 5—figure supplement 2*), corroborating our findings on Krt15 protein levels (*Figure 4A*). On the other hand, genes for various mitochondrial proteins, involved in mitochondrial energetics, redox or biogenesis, were upregulated particularly in the stronger Drp1 knockdown (*Figure 5B* upper panel, *Figure 5—figure supplement 1C* (bold), *Figure 5—figure supplement 2B*). This is consistent with our observation of decreased mitochondrial content in the weaker Drp1 knockdown (*Figure 4J*). Synthesis of the mitochondrial proteins can be potentially sustained by elevated ribosomal genes and genes involved in protein synthesis in the stronger Drp1 knockdown (*Figure 5B*, lower panel). This finding is consistent with observations linking Drp1 repression to ribosomal genes and/or protein synthesis (*Tanwar et al., 2016*; *Zhao et al., 2021*; *Favaro et al., 2019*). We speculate that repression of Drp1 sustained beyond an optimal level or time may initiate compensatory changes to trigger mitochondrial biogenesis through retrograde signaling (*Ryan and Hoogenraad, 2007*), which may be detected beyond ~50 % reduction of Drp1 protein when transformation efficacy starts declining (*Figure 4—figure supplement 1A*).

Cell clustering in a UMAP plot (*Figure 5—figure supplement 1D*, *Figure 5—source data 1*) revealed remarkable correspondence to the conserved cluster markers of the Parental/transformed data set (*Figure 2*, *Figure 5—figure supplement 1D,E*), allowing assignment of cluster identities of G2-M, S-early, S-late, mtDNA-lo, SloCycl-K15-hi and Cytokine to the control and Drp1 knockdown data set. Strikingly, the weaker Drp1-knockdown population exhibits expansion of the SloCycl-K15-hi cluster and reduction of the Cytokine and mtDNA-lo clusters observed in the stem/progenitor cell enriched Drp1-repressed-TF-1 population, albeit to a lesser degree (*Figure 5C*). This similarity in cluster organization suggests that fine-tuned Drp1 repression in Parental population establishes a stem/progenitor cell-like gene expression profile to poise them for accelerated neoplastic transformation (*Figure 4D*). On the other hand, the stronger Drp1 knockdown markedly reduces the abundance of the SloCycl-K15-hi cluster and exhibits overall opposite trend of cluster profile compared to the weaker Drp1 knockdown and TF-1 population (*Figure 5C*). *KRT15* and various other marker Keratin genes are dramatically suppressed across the clusters in the stronger Drp1 knockdown and overall elevated in the weaker Drp1 knockdown (*Figure 5—figure supplement 1F*), confirming and expanding our findings on Krt15 protein levels (*Figure 4A*). Moreover, analyses of cell cycle distribution (as in *Figure 2*) revealed higher G2-M score, reduced S score and perturbed G1 to S reciprocity between SloCycl-K15-hi and the mtDNA-lo clusters particularly in the stronger Drp1 knockdown (*Figure 5D*, *Figure 5—figure supplement 1G*). The dramatic reduction of the SloCycl-K15-hi cluster in the stronger Drp1 knockdown cells could be due to their inability of exiting the elevated CyclinB1/CDK1-driven G2-M regulation in this cluster (*Figure 5E*, left). Also, the stronger Drp1 knockdown maintained marked elevation of the growth factor cyclin, Cyclin D2 and reduction of the S phase marker, PCNA, across clusters (*Figure 5E*, right); Cyclin E1 and Sox2 transcripts were not detected in this data set. Therefore, our data demonstrates that particularly fine-tuned Drp1 repression sustains a gene expression profile characteristic of the neoplastic stem/progenitor cell enriched Drp1-repressed-[CyclinE$^{hi}$-Sox2$^{hi}$-Krt15$^{hi}$]-TF-1 population, which is prevented by more complete Drp1 repression that induces cell cycle disturbances.

Taken together, single-cell analyses of stem/progenitor cell regulators (using immunocytochemistry and transcriptomics) and mitochondrial shape (using quantitative [Fission]/[Fusion] metrics) of our newly derived transformed keratinocyte model reveal that a slow cycling 'stem/progenitor-like [CyclinE$^{hi}$-Sox2$^{hi}$-Krt15$^{hi}$] state' is primed by a 'goldilocks' level of fine-tuned Drp1 repression (Drp1$^{ftr}$) that maintains small networks of fused mitochondria (*Figure 5F*). The stem/progenitor state driven neoplastic transformation is accelerated by fine-tuned Drp1 repression maintained by reduced Drp1 protein levels, while the neoplastic stem/progenitor state is sustained by fine-tuning Drp1 by reducing its S-616 phosphorylation that modulates mitochondrial potential. Given the bidirectional crosstalk of mitochondrial dynamics and redox status (*Willems et al., 2015*), we speculate that fine-tuned Drp1 repression driven stem/progenitor cell priming may involve sustaining their nuanced redox environment (*Bigarella et al., 2014*). The Drp1 repression-driven CyclinE$^{hi}$ state may allow rapid transition from quiescence to a self-renewing/proliferation state to support neoplastic transformation (*van Velthoven and Rando, 2019*), given Cyclin E is indispensable for exiting quiescence or for neoplastic transformation (*Hwang and Clurman, 2005*; *Liu et al., 2019*; *Teixeira and Reed, 2017*).

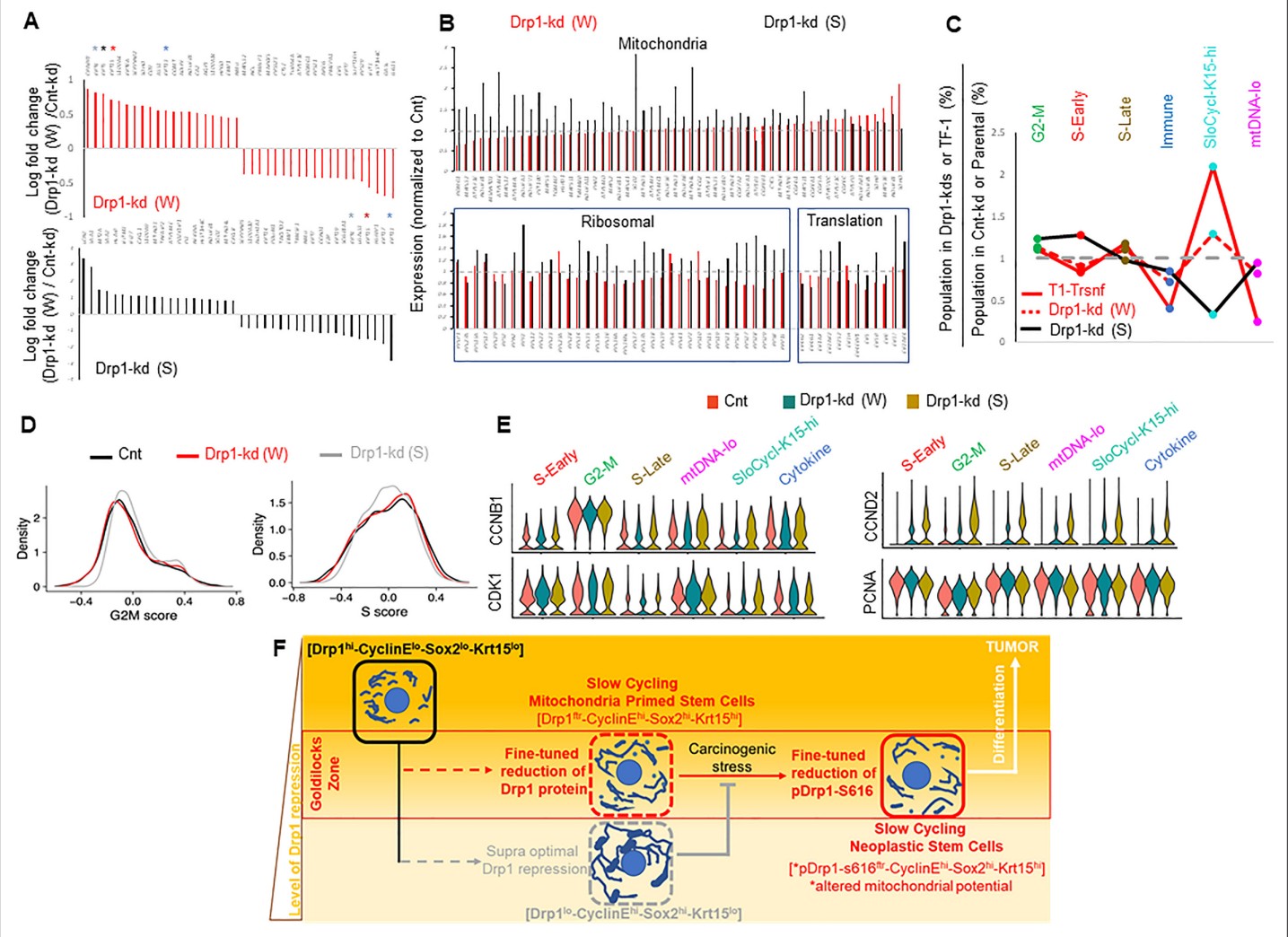

**Figure 5.** Fine-tuned Drp1 repression sustains transcriptomic profile similar to the neoplastic stem/progenitor cell enriched TF-1 population, which is suppressed by more complete Drp1 repression. (**A**) Bar plots showing positive and negative log fold change values of top 20 genes in Parental cells with weaker (**W**) (Top) or stronger (**S**) (bottom) Drp1 knockdown with respect to control knockdown; 0 signifies no change and color coded * denote relevant *KRT* genes (larger figure in *Figure 5—figure supplement 2A*). (**B**) Bar plot showing ratio of absolute expression values of mitochondrial genes (top) and ribosomal or protein translation genes (bottom) in the Parental cells with Drp1 knockdown normalized to control knockdown; one signifies no change (larger figure in *Figure 5—figure supplement 2B*). (**C**) Line plot showing % population of Parental cells with Drp1 knockdown in the named scRNA-seq derived cell clusters, with respect to control knockdown. This is compared to the same in the TF-1 cells with respect to its control Parental population; gray dashed line signifies no change from respective controls. (**D**) Distribution of G2-M and S scores obtained from gene expression analyses of cell cycle predictive genes in Parental cells with Drp1 or control knockdown. (**E**) Violin plots depicting expression of key cell cycle genes in the designated scRNA-seq derived cell clusters in Parental cells with Drp1 or control knockdown. (**F**) Working model describing the key findings: a 'goldilocks zone' of fine- tuned Drp1 repression, achieved by reduction of protein levels or its S616 phosphorylation, maintains smaller fused mitochondrial networks and primes a stem/progenitor cell state to support carcinogenic transformation; more complete Drp1 repression sustains hyperfused mitochondria and prevents stem/progenitor cell state dependent neoplastic transformation.

The online version of this article includes the following source data and figure supplement(s) for figure 5:

**Source data 1.** Cluster markers identified in Parental HaCaT cells expressing non-targeting, strong or weak Drp1 shRNA.

**Figure supplement 1.** Further scRNA-seq analyses of Parental HaCaT cells expressing non-targeting shRNA, strong or weak Drp1 shRNA.

**Figure supplement 2.** Enlarged *Figure 5A and B* with legible gene names.

Such transition may involve further activation of mitochondrial fission induced by signaling pathways supporting stemness (*Spurlock et al., 2019*), which would be prevented by more complete Drp1 repression (*Figure 5F*) or total ablation of Drp1 (*Kashatus et al., 2015*; *Serasinghe et al., 2015*). This conceptualization helps explain how regulated reduction and increase of Drp1 activity can potentially

maintain adult stem cell properties (*Iwata et al., 2020*; *Khacho and Slack, 2017*; *Parker et al., 2015*; *Xie et al., 2015*). Further in vivo validation of our findings would reveal if and how bodily stresses causing mild repression of Drp1-driven mitochondrial fission can possibly sustain the mitochondria primed 'stem/progenitor-like state' in tissues making them susceptible to neoplastic transformation by chemicals and/or genetic changes. Enhanced mitochondrial fusion, proposed as a required event of neoplastic transformation of stem cells (*Bonnay et al., 2020*), may also involve such mitochondria-driven priming.

# Materials and methods

**Key resources table**

| Reagent type (species) or resource | Designation | Source or reference | Identifiers | Additional information |
|---|---|---|---|---|
| Strain, strain background (Mouse, female) | Athymic Nude mice | Jackson Lab | RRID:IMSR_JAX:002019 | |
| Cell line (Human) | HaCaT | M.Athar PMID:27725709 | Gift | Originally obtained from Addex BioTechno-logies |
| Recombinant DNA reagent | pBABE-hDrp1 (plasmid) (Human) | PMID:25658204 | Gift | Dr. Jerry ChipuK Lab |
| Recombinant DNA reagent | pBABE-hDrp1S616A (plasmid) (Human) | PMID:25658204 | Gift | Dr. Jerry ChipuK Lab |
| Recombinant DNA reagent | pBABE-hDrp1S616D (plasmid) (Human) | PMID:25658204 | Gift | Dr. Jerry ChipuK Lab |
| Recombinant DNA reagent | Human *DNM1L* shRNA1 (plasmid) | Dharmacon PMID:26446260 | Dharmacon Clone: TRCN0000001099 | Our previous publication |
| Recombinant DNA reagent | Human *DNM1L* shRNA2 (plasmid) | PMID:26446260 | Dharmacon Clone: TRCN0000001097 | Our previous publication |
| Recombinant DNA reagent | Non-targeted shRNA control (plasmid) | PMID:26446260 | Sigma SHC016 | Our previous publication |
| Recombinant DNA reagent | pCMX-mito-PSmOrange (fluorescent protein reporter) | PMID:30910831 | Available from us on request. | Our previous publication |
| Antibody | Drp1 (Mouse, Clone 8/DLP1) | BD Transduction | 611,112 RRID:AB_398423 | WB (1:1000) IF (1:100) |
| Antibody | pDrp1 S616 (Rabbit, clone D9A1) | Cell Signaling | 4,494 RRID:AB_11178659 | WB (1:1000) IF (1:100) |
| Antibody | Cyclin Sampler (Mouse or Rabbit monoclonal) | Cell Signaling | 9,869 RRID:AB_1903944 | WB (1:1000) IF (1:100) |
| Antibody | Actin (Mouse, clone Ab5) | BD Transduction | 612656 RRID:AB_2289199 | WB (1:10000) |
| Antibody | Sox2 (Mouse, clone 030–678) | BD Pharmingen | 561,469 RRID:AB_10694256 | WB (1:500) IF (1:100) |
| Antibody | HSP60 (Mouse, clone 24/HSP60) | BD Transduction | 611,563 RRID:AB_399009 | IF (1:200) |
| Antibody | pCyclin E- T62 (Rabbit polyclonal) | Cell Signaling | 4,136 RRID:AB_2071080 | IF (1:50) |
| Antibody | Fis1 (Mouse, clone B-5) | SantaCruz | sc-376447 | WB (1:1000) |
| Antibody | TOM20 (Mouse, clone D8T4N) | Cell Signaling | 42,406 RRID:AB_2687663 | WB (1:5000) IF (1:200) |

*Continued on next page*

*Continued*

| Reagent type (species) or resource | Designation | Source or reference | Identifiers | Additional information |
|---|---|---|---|---|
| Antibody | MFN2 (Rabbit, clone 7H42L13) | ThermoFisher | 702,768 | WB (1:1000) |
| Antibody | MFN1 (Rabbit, clone D6E2S) | Cell Signaling | 14,739 RRID:AB_2744531 | WB (1:1000) |
| Antibody | Opa1 (Mouse, clone 1E8-1D9) | ThermoFisher | MA5-16149 RRID:AB_11153569 | WB (1:1000) |
| Antibody | AHR (Mouse, clone RPT1) | ThermoFisher | MA1-514 RRID:AB_2273723 | WB (1:1000) |
| Antibody | CytochromeC (Rabbit polyclonal) | Cell Signaling | 4,272 RRID:AB_2090454 | WB (1:1000) |
| Antibody | Cytokeratin15 (Rabbit, clone EPR1614Y) | Abcam | ab52816 RRID:AB_869863 | WB (1:2000) IF (1:200) |
| Chemical compound, drug | TCDD | Sigma | 48,599 | Solution in Toluene |
| Software, algorithm | MitoGraph v2.1 | PMID:30910831 PMID:25640425 | https://github.com/vianamp/MitoGraph (*Viana, 2020*) | Rafelski lab; freely available |
| Chemical compound, drug | Fluoromount G | Southern Biotech | 0100–01 | |
| Other | FuGENE 6 (Transfection Reagent) | Promega | E2691 | |
| Other | Lipofectamine 3,000 (Transfection Reagent) | Invitrogen | L3000001 | |
| Other | sc-RNAseq data generated here in | This paper | Gene expression omnibus GSE171772 | Raw, analyzed and meta data deposited in Gene expression omnibus |
| Software, algorithm | Cell Ranger | 10 X Genomics | RRID:SCR_017344 | |
| Software, algorithm | Seurat | https://satijalab.org/seurat/get_started.html | RRID:SCR_016341 | Freely available |

## Cell culture methods

HaCaT cells were cultured on plastic dishes in Dulbecco's Modified Eagle's Medium (DMEM) with high glucose (4.5 g/L), sodium pyruvate (1 mM), L-Glutamine (4 mM), Penicillin (100 U/mL), Streptomycin (100 µg/mL), and 10 % FBS using standard techniques. Transformed HaCaT cells were derived from the Parental HaCaTs and HaCaTs stably expressing mito-PSmO (and mito-roGFP) developed previously (*Spurlock et al., 2019*) by selecting multiple clones after treatment with the noted dose of TCDD with replenishment with fresh TCDD every 2–3 days. The key difference in cell proliferation and cell morphology between TF-1 and TF-10 populations have been confirmed in multiple cell transformation experiments.

Transfections were preformed using Fugene six or Lipofectamine 3000, appropriately modifying manufacturer's protocols. When required, lines stably expressing non-targeted human *DNM1L* shRNAs, Drp1-WT or S-616A mutants, or mito-PSmO were selected using puromycin treatment (2 µg/mL) with regular media changes for 2 weeks until resistant colonies grew. Any stable genetically modified or transformed line and their parental line was treated with two courses of BM-Cyclin treatment to eliminate any mycoplasma infection.

Standard cell proliferation assay was performed as described in *Parker et al., 2015*. The extreme limiting dilution analysis (ELDA) of spheroid formation ability was performed by seeding 1, 10, 100, and 1000 cells per well with 24 wells per dose in a 96-well UltraLow Attachment plate and TIC-supporting media (DMEM:F12 supplemented with human growth factors as published elsewhere *Spurlock et al., 2019*). Wells were Supplemented with media equaling 10 % of the total volume every second day. On

Day 5, the total number of wells for each dose containing nascent spheroids was tabulated. The web ELDA tool was used to analyze the results (*Hu and Smyth, 2009*).

## Immunoblotting and immunofluorescence

Immunoblotting was performed using standard techniques. Equal cell equivalents of whole cell lysates (assessed by counting a fraction of the cell suspension) were run on 10 % polyacrylamide gels and transferred to PVDF membranes followed by probing with appropriate primary and secondary antibodies. Band intensity was quantified using ImageJ and normalized by loading and experimental controls as indicated.

Immunofluorescence was performed as described previously (*Parker et al., 2015*) on cells seeded in Nunc Lab-Tek chamber slides. Briefly, cells were fixed in freshly prepared aqueous paraformaldehyde (4%) supplemented with (4%, w/v) sucrose and permeabilized in freshly prepared Triton X-100 (0.1%) prior to staining. Immunostained cells were mounted in FluoromountG with Hoechst 33,342 (10 μg/mL).

## Confocal microscopy, image processing and analysis

Confocal microscopy was performed on a Zeiss LSM700 microscope equipped with 405 nm, 488 nm, 555 nm, and 639 nm lasers, using the proprietary Zen Black (2012) software. Confocal micrographs were acquired with optimized laser powers and appropriate filters to minimize crosstalk, cross-excitation and bleaching. Live cells were imaged using a temperature and $CO_2$ controlled chamber set to 37 °C and 5 % $CO_2$. Mitochondrial potential per unit mass was determined from TMRE and Mitotracker green co-stained cells as described previously (*Mitra and Lippincott-Schwartz, 2010*).

Image processing and analysis of relative protein abundance and localization were performed using Zen Black and Zen Blue software to obtain background corrected mean fluorescence intensities within defined regions of interest drawn on maximum intensity z-projections of optical slices. Nuclear and cytosolic regions were demarcated based on DNA stain Hoechst 33,342 and the immunostain of the molecule of interest, respectively. The [Fission] and [Fusion1/5] metrics were computed using MitoGraph v2.1 software run on 3D stacks of confocal optical slices acquired from live cells stained with Mitotracker Green, as described previously (*Spurlock et al., 2019*; *Spurlock and Mitra, 2021*). [Fission] is total mitochondria number / total mitochondrial length; [Fusion5] is (sum of top five mitochondrial length / total mitochondrial length) X 100. Mitochondrial matrix continuity was assessed from time lapse images of mito-PSmO-expressing cells after photo-conversion with a 488 nm laser within a 50 × 50 px region of interest, as described previously (*Spurlock et al., 2019*; *Spurlock and Mitra, 2021*). Microsoft Excel and IBM SPSS Statistics 23 were used to perform computations, bivariate analyses and significance testing.

## Flow cytometry with TMRE

Cells were trypsinized and stained in suspension with TMRE and processed as described previously (*Spurlock et al., 2021*). Data from ~10,000 single cells were acquired using PE-Green set-ups on the LSR II flow cytometer (BD Biosciences), after gated by forward and side scatter. Analyses was done using FlowJo software using the unstained cells as controls, and histograms of TMRE fluorescence are reported here.

## Tumor forming assay in mice and histochemistry

We first performed a pilot experiment to determine the number of cells and time required to form xenografts with our newly derived transformed lines, using established cancer line as positive control. Thereafter, we performed the second experiment with eight animals per group, as determined based on various prior publication by our group and others. Our data includes both experiments. Five million transformed HaCaT cells were injected subcutaneously in the flank of athymic nude mice to form xenograft tumors. Prior to injection, cells were trypsinized and washed once with sterile PBS, and finally resuspended to 25,000 cells/μL in room temperature PBS with Geltrex. Tumors arising within 6–8 months were harvested for histochemical analyses. The harvested tumors were frozen unfixed in TissueTek OCT compound on dry-ice. A Cryostat was used to obtain 5 μm slices, which were then fixed using freshly made cold 4 % paraformaldehyde and then stained with H&E using standard techniques.

## Single-cell RNAseq and data analyses

Trypsinized single cells were washed in PBS (without ca ++ or Mg++) with 0.04 % BSA and tested for >90% viability. Single-cell analysis was performed on a 10xGenomics platform according to 3' v3.1 NextGem Dual Index manual. The 3'-biased cDNA libraries were constructed through the following steps: cDNA fragmentation, end repair and A-tailing, and size selection by SPRIselect beads, adaptor ligation, and sample index PCR amplification, and then SPRIselect beads size selection again. The final constructed 3'-biased single-cell libraries were sequenced by Illumina Nextseq500 machine, targeting total reads per cell for 20,000 at minimum, and the sequencing cycles consisted of 28 bp for read 1, 90 bp for read 2, and 10 bp for i7 index, and 10 bp for i5 index.

Count matrices were generated from the single-cell raw fastq files using 10 x genomics *cellranger* software (v.4.0) using hg38 reference genome provided by 10 x Genomics. Raw, analyzed and meta data are available in Gene expression omnibus (GEO GSE171772). We filtered the data to only include cells expressing at least 2000 detectable genes (using 'nFeature_RNA' filter); this also filtered out dead cells with >10% mt-DNA gene expression in our data set. Our data coverage included approximately ~3000 genes in average of 5000 cells in each population. The resulting count matrices were analyzed by Seurat package (v3.9.9) with the standard workflow (*Butler et al., 2018*; *Stuart et al., 2019*). After applying the (*nFeature_RNA* >2000), the expression data were then normalized using the *NormalizeData* function in Seurat and variable featuresets were identified using the *FindVariableFeatures* function in Seurat. Depending on the type of comparison, specific sets of samples were combined using the *IntegrateData* function. The integrated datasets were scaled and cells were clustered with up to 20 dimensions (dim = 1:20). We performed clustering using several resolution parameters (0.1–0.8) and visually selected optimal resolution for specific datasets. The clusters were visualized using UMAP with up to 20 dimensions. Marker genes for each cluster were calculated with the *FindAllMarkers* function and statistical significance was calculated by Wilcoxon Rank Sum test. Differential expression of the cluster markers was carried out using the FindMarkers function of Seurat. For this, we only used the normalized data before integration (assay 'RNA') using the Wilcoxon rank sum test, as suggested by Seurat documentation. For cell-cycle scoring, we use the CellCycleScoring function of Seurat, which provides scores of G2M and S phase scores and assigns cell-cycle phases based on G2-M and S scores, or in G1/G0 when both scores are low (*Tirosh et al., 2016*). G1 to S reciprocity was obtained as: {% [G1/S] in Cluster 3 / % [G1/S] in Cluster 5}. The cell-lineage determination and trajectory calculation were carried out using the Slingshot (*Street et al., 2018*) package with Seurat clusters and default parameters. The algorithm was allowed to automatically identify the start and the end clusters. GSEA analyses were performed with ranked log fold change (LFC) of markers ( > 0.1, p < 0.01, FDR q value p < 0.01) using MsigDB as described in *Tanwar et al., 2016*. Overrepresentation analyses were performed with genes with negative LFC in Cluster 5 using the Reactome PA package of R v 3.6.0.

## Statistical analyses and replicates

Drawing of plots and statistical analyses were performed using Excel, SPSS or R package, as appropriate. For the cytochemistry-based dot-plot analyses, the following operations were performed: signal threshold = [Median] + 1.5 X [MAD], where Median Absolute Deviation (MAD) is the median of the absolute difference of individual value from the population median. Nonetheless, the reduced quantitative nature of immunocytochemistry studies compared to immunoblotting studies is exemplified in the smaller increase in Sox2 in the TF-1 cells in immunocytochemistry (*Figure 3B, C and G*) in comparison to the ~8 -fold increase observed in immunoblot (*Figure 3A*).

Experiments have been performed multiple times with separate culture sets to serve as biological replicates, sometimes by different personnel to ensure reproducibility. Key findings have also been cross validated by other relevant experiments when appropriate. In majority of the cases, 2 or more technical replicates for each biological replicate have been used with positive and/or negative controls, as appropriate. The key findings of the scRNA-seq data has been validated in multiple independent experiments. The high-resolution confocal micrograph analyses using Mitograph software involve the quality control of manual exclusion of cells where digital images do not match the corresponding confocal micrograph (*Spurlock and Mitra, 2021*). Without this appropriate quality control step the data may remain erroneous.

## Acknowledgements

BS, KM, AM are supported by NIH award (# R33ES025662). We acknowledge Dezhi Wang from the Pathology Core Research Laboratory for preparation of H&E stained slides, UAB Flow Cytometry and Single Cell Services Core (supported by P30 AR048311, P30 AI027667) and the Dr. Mike Crowley from the Heflin Genomics Core for scRNA-seq, and Dr. J Chipuk for the Drp1 constructs, and Dr. M Athar for the HaCaT cell line.

## Additional information

### Funding

| Funder | Grant reference number | Author |
| --- | --- | --- |
| National Institute of Environmental Health Sciences | R33ES025662 | Kasturi Mitra Brian Spurlock Aida Moran |
| National Institutes of Health | P30 AR048311 | Shanrun Liu |
| National Institutes of Health | P30 AI027667 | Shanrun Liu |

The funders had no role in study design, data collection and interpretation, or the decision to submit the work for publication.

### Author contributions

Brian Spurlock, Data curation, Formal analysis, Investigation, Methodology, Validation, Visualization, Writing - review and editing; Danitra Parker, Data curation, Methodology, Resources; Malay Kumar Basu, Formal analysis, Methodology, Software, Writing - review and editing; Anita Hjelmeland, Gene P Siegal, Methodology, Writing - review and editing; Sajina GC, Shanrun Liu, Methodology; Alan Gunter, Data curation, Formal analysis; Aida Moran, Data curation; Kasturi Mitra, Conceptualization, Formal analysis, Funding acquisition, Methodology, Supervision, Writing - review and editing

### Author ORCIDs

Brian Spurlock http://orcid.org/0000-0002-9757-4494
Sajina GC http://orcid.org/0000-0003-2676-2794
Kasturi Mitra http://orcid.org/0000-0003-3718-7094

### Ethics

This study was performed in strict accordance with the recommendations in the Guide for the Care and Use of Laboratory Animals of the National Institutes of Health. All of the animals were handled according to approved institutional animal care and use committee (IACUC) protocols (IACUC-21347) of the University of Alabama at Birmingham. All mice were examined daily, and were euthanized by $CO_2$ inhalation, and death confirmed by cervical dislocation, as approved by the IACUC protocol.

### Decision letter and Author response

Decision letter https://doi.org/10.7554/eLife.68394.sa1
Author response https://doi.org/10.7554/eLife.68394.sa2

## Additional files

### Supplementary files

• Transparent reporting form

### Data availability

Details pertaining to the sc-RNAseq experiment are available as Supplementary data Tables 1, 2, 3. Raw, analyzed and meta data are available in Gene expression omnibus (GEO GSE171772).

The following dataset was generated:

| Author(s) | Year | Dataset title | Dataset URL | Database and Identifier |
|---|---|---|---|---|
| Spurlock B, Basu MK, Mitra K | 2021 | Fine-tuned repression of Drp1 driven mitochondrial fission primes a 'stem/progenitor-like state' to accelerate neoplastic transformation | https://www.ncbi.nlm.nih.gov/geo/query/acc.cgi?acc=GSE171772 | NCBI Gene Expression Omnibus, GSE171772 |

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
