## [Decision Letter]

**Acceptance summary:**

In their paper, Spurlock and colleagues look at the role of mitochondria fusion caused by Drp1 repression in driving the stem/progenitor-like state of skin stem cells. Prior work hinted at the possibility that mitochondrial fission/fusion activity is important in supporting neoplastic transformation, but it was unclear exactly what this role was. Here, the authors use an assay for neoplastic transformation induced by carcinogen treatment to demonstrate that diminution in mitochondrial fission activity (from increased phosphorylated Drp1 pools) can prime a stem/progenitor-like state in carcinogen-treated cells, leading to accelerated neoplastic transformation. Using genetic strategies and single cell RNAseq they additionally show that only partial repression of Drp1 is necessary for establishing the stem/progenitor-like state for driving neoplastic transformation, with too much or too little Drp1 repression having no effect. The data are therefore relevant for understanding the conditions for driving neoplastic transformation. The work helps to clarify the mitochondria's role in neoplastic transformation.

**Decision letter after peer review:**

Thank you for submitting your article "Fine-tuned repression of Drp1 driven mitochondrial fission primes a 'stem/progenitor-like state' to accelerate neoplastic transformation" for consideration by *eLife*. Your article has been reviewed by 3 peer reviewers, and the evaluation has been overseen by Utpal Banerjee as the Reviewing and Senior Editor. The reviewers have opted to remain anonymous.

All three reviewers have agreed that your paper will be acceptable for *eLife*, if their concerns are met. Please note that most of the required changes below are for presentation and quality. But this is important and please do attend to these suggestions.

Essential revisions:*Reviewer 1.*

1. The abstract is too vague and unstructured for readers to understand what the paper is about and needs significant improvement.

2. The introduction does not clearly define the problem being addressed in the paper. Among the issues is whether the study is simply about changes in Drp1 levels and its effects on neoplastic transformation, or whether it is focused on mitochondria fission/fusion and its impact on neoplastic transformation. If the latter is part of the conclusion, then the authors need some other way to manipulate mitochondrial fission/fusion levels other than using Drp1 (i.e., changing levels of Opa1, mitofusins, etc) to see whether this also can impact neoplastic transformation.

3. It would be nice to see if the membrane potential of mitochondria differs under TF1- versus TF10- treatment. As it now stands, all the authors provide is a visual inspection of mitochondria under these two conditions, providing unclear metrics for determining the degree of fragmentation or fusion.

4. Given their speculation about cyclin E, *Sox2* and Krt15 levels being high in the TF1 treated cells, can the authors use antibodies against these proteins to prove it, showing staining is higher in TF1-treated cells compared to TF10 treated cells? In addition, do the authors see more cyclin E associated with mitochondria during TF1 treatment relative to TF10 treatment?

5. The authors should provide a better discussion to help readers understand the significance of their results.

*Reviewer 2.*

1. This paper relies almost entirely on a single derivative of a single cell line. The authors should validate their claims with independent corroboration, using other cells and/or other derivatives.

(Editor's comment: We understand that repeating all experiments with a second cell line will pose a big challenge. In consultation with the referees, addressing this particular concern is optional and left up to the author's discretion.)

2. Additional weaknesses include the reliance on a single carcinogen and on a single blot for expression of the fission and fusion proteins.

(Editor's comment: likewise, repeating all experiments with a second carcinogen and blot will pose a big challenge. In consultation with the referees, addressing this particular concern is optional and left up to the author's discretion. However, do address the reason why this concern was raised (below in 8))

3. This is a concern because the increased levels of Drp1 in the TF1 cells in Figure 1D conflicts with what appears to be an overall decrease in Drp1 levels in TF1 in Figure 3C.

4. The authors use an unusual method to characterize mitochondrial dynamics:

Explain why other existing methods for quantification that give more meaningful numbers and are more readily understood by readers were not used.

Is it also possible that a change in fission rates is balanced by a concurrent change in fusion rates, as suggested by the increase in Drp1 and Mfn1 levels in Figure 1C. The authors should address this, since so much of the paper rests on the assumption that fission rates are decreased in a subpopulation of cells.

(Editor's comment : Address this in the text, no new experiments needed)

*Reviewer 3.*

1. Quantification of immunoblots in Figures 1D, 3E, 3F: Where the immunoblots are used to make arguments about increases and decreases in key proteins such as pDrp1 or Krt15 levels, these must be quantified in some manner.

(Editor's comment : to avoid delays in publication, this is optional if it will require new experiments, but do quantitate existing blots if possible and address the issues raised)

2. The placement of the arrow in 2B is quite confusing and doesn't seem to indicate the main point trying to be made which is more about Cluster 3 and Sox4 and Krt15 than Krt13 which doesn't seem to change.

3. Quantification of Dot plots: While the dot plots are helpful for seeing differences between individual cells within the populations, several conclusions are based on the interpretation of how many cells from one condition vs another are within a certain range or quadrant but these are not quantified. For example, 3B what percent of each condition are in each quadrant? This is hard to see by eye. Similarly, 3C, 3D, 3I, and 4B must be quantified and some measure of statistical significance presented. And what is the p-value shown in 4A referring to? Differences between all of them or individual comparisons?

4. The resolution of figures 4D and 4E must be improved so that the gene names can be seen upon zooming in.

5. 4C: Is the difference in fusion1 statistically significant? how do the other populations look in the fission vs fusion1 plot?

6. Ext Figure 2: 2a: may be better as a table instead of a graph because the graph is hard to read and compare. 2c-2d: Cluster 5 seems to be de-enriched for many genes so is it truly surprising or significant that there is less mtDNA? Are the overall counts lower? How are the data normalized to account for differences?

7. Extended Figure 4A (in figure legend or text please indicate the actual efficiency values and show error bars for replicates as the level of Drp1 repression/kd efficiency is a vitally important metric).

8. No key in extended Figure 4F.

9. Materials and methods: There are several key details missing such as the identity (manufacturer) and concentration of all of the antibodies used in the immunoblots and immunofluorescence analysis such as those for pDrp1, Opa1, Mfn1, Mfn2, Cyclins E2, A2, B1, D2, Cytochrome C, Fis1, Tom20. These details are not only important for reproducibility but as well as for evaluation of the results by experts familiar with these particular reagents.

a. For immunoblotting of whole cell lysates, how did the authors make sure equivalent amounts of cells were analyzed, especially given their very different proliferation characteristics?

b. How many replicates were used for the single cell RNASeq analysis of TF-1/TF-10 and of weak/strong Drp1-shRNA? What was the difference between the replicates especially for the cluster analysis as done in figure 2A (can this be shown in supplemental material?) to get an idea of how reproducible the findings are especially given how such a small difference in Drp1-shRNA induced levels seems to have such a striking difference on gene expression.

c. Additionally, there are several typos in this section (p22 Lines 479, 481, 491. p24 lines 524, 525, 529).

---

## [Author Response]

Essential revisions:Reviewer 1.1. The abstract is too vague and unstructured for readers to understand what the paper is about and needs significant improvement.

We thank the Reviewer for this comment. In the revised manuscript, we have made the abstract more succinct. The revised abstract also includes the clarification based on new experimental results: fine-tuned reduction of Drp1 protein level accelerates stem/progenitor cell driven neoplastic transformation, while the neoplastic stem/progenitor cell markers are maintained by reduction of the activating phosphorylation of Drp1.

2. The introduction does not clearly define the problem being addressed in the paper. Among the issues is whether the study is simply about changes in Drp1 levels and its effects on neoplastic transformation, or whether it is focused on mitochondria fission/fusion and its impact on neoplastic transformation. If the latter is part of the conclusion, then the authors need some other way to manipulate mitochondrial fission/fusion levels other than using Drp1 (i.e., changing levels of Opa1, mitofusins, etc) to see whether this also can impact neoplastic transformation.

We thank the Reviewer for this comment. We have now clearly defined our focus on Drp1 in the abstract and in the last paragraph of the Introduction section (Lines: 89-98). There, we also clarify that our self-renewing/proliferating cells enriched model is characterized by robust repression of Drp1 activity in comparison to the core regulators of mitochondrial fusion (Figure 1D). Therefore, we focused on Drp1 driven mitochondrial shape and functional changes. Both fusion and fission processes are mentioned in the previous paragraphs of Introduction since they oppose each other.

3. It would be nice to see if the membrane potential of mitochondria differs under TF1- versus TF10- treatment. As it now stands, all the authors provide is a visual inspection of mitochondria under these two conditions, providing unclear metrics for determining the degree of fragmentation or fusion.

We thank the Reviewer for this comment. We now demonstrate that the mitochondrial structural properties in the TF-1 cells is associated with overall lowering of mitochondrial transmembrane potential in comparison to Parental or the TF-10 cells, assessed by their TMRE uptake (Figure 1F and its legend, Lines: 166-171). Furthermore, the mitochondrial potential per unit mitochondrial mass in individual cells (TMRE / Mitotracker Green staining) is regulated over a wider range particularly in the TF-1 cells, while it is lowest in the TF-10 cells (Figure 1—figure supplement 1G and its legend). Also, expression of the phospho-deficient Drp1-S616A mutant in TF-1 cells maintains markedly lower mitochondrial membrane potential in comparison to the Drp1-WT expression (Figure 3J and its legend, Lines: 348-356), suggesting reduced pDrp1-S616 levels maintains modulates mitochondrial potential in the TF-1 cells. Notably, such impact of Drp1-S616A expression on mitochondrial potential is not observed in the Parental cells (Figure 3J and its legend), where pDrp1-S616 levels and mitochondrial membrane remain higher than TF-1 cells. Notably, the accelerated carcinogen driven neoplastic transformation caused by Drp1 knockdown does not involve modulation of mitochondrial potential (Figure 4H and its legend, Lines: 407-409). The relevant methods are described in Lines: 593-94, 610-615.

Clarification of the ‘unclear’ metrics for determining the degree of fragmentation or fusion:

We designed, validated (using Drp1 and mitofusin knockouts) and demonstrated the utility of the mitochondrial [Fission] and [Fusion] metrics (as a part of the mitoSinCe^2^ assay and described in Methods, Lines: 600-604) towards filling a gap in quantitation of mitochondrial structure/morphology. As described in our original publication, these metrics quantify the ‘contribution’ of the dynamic fission and fusion processes to the steady state mitochondrial shape in single cells, and do not quantify kinetics of mitochondrial fission or fusion processes. The main advantage of the quantitative mitochondrial [Fission] and [Fusion] metrics is that they together can identify heterogeneity in mitochondrial structure in a given cell population (Figures 1E, 4I), which other traditional methods cannot. Nonetheless, we have now added analyses of the standard photoconversion based matrix continuity assay confirming the Figure 1D predicted higher mitochondrial matrix continuity in TF-1 and TF-10 cells in comparison to the Parental cells (Figure 1—figure supplement 1D). We have clarified the above points in the revised manuscript (Lines: 136-165). (Please see response to point 9 for further details).

4. Given their speculation about cyclin E, Sox2 and Krt15 levels being high in the TF1 treated cells, can the authors use antibodies against these proteins to prove it, showing staining is higher in TF1-treated cells compared to TF10 treated cells? In addition, do the authors see more cyclin E associated with mitochondria during TF1 treatment relative to TF10 treatment?

Our conclusions about Cyclin E, *Sox2* and Krt15 being high in the TF1 cells are based on immunoblotting and immunocytochemistry studies of these proteins (Figure 3A-G), which validate the scRNAseq data (Figure 2B, except Cyclin E that was not covered reasonably well in our data set).

We thank the Reviewer for the query on the mitochondrial Cyclin E pool that we reported previously. Indeed, majority of the cytosolic Cyclin E particularly in the Drp1 knockdown cells represent mitochondrial Cyclin E (not shown), which remains to be elucidated further (Lines: 3-383-385). We are currently pursuing this line of investigation in detail and therefore left out of this short report.

5. The authors should provide a better discussion to help readers understand the significance of their results.

We thank the reviewer for this comment. We have now expanded our Discussion section along with each result section (as required for a short report format) and also provided a schematic model for better understanding of the conclusions (Figure 5F, Lines: 518-530).

Reviewer 2.1. This paper relies almost entirely on a single derivative of a single cell line. The authors should validate their claims with independent corroboration, using other cells and/or other derivatives.(Editor's comment : We understand that repeating all experiments with a second cell line will pose a big challenge. In consultation with the referees, addressing this particular concern is optional and left up to the author's discretion.)

We thank the Editor and the Reviewers for understanding that repeating experiments in yet another cell line is a big challenge. We also reason this effort is unnecessary because we used the HaCaT cell line and its new derivatives as a model to establish the main conclusions. As highlighted by other Reviewers, we arrived at our main novel conclusion from in depth multidimensional characterization of this in vitro model system, and now we have provided further characterization. Moreover, our finding that close to complete knockdown of Drp1 prevents neoplastic transformation is consistent with the findings from other labs in other cell models and also raises the possibility that inability to sustain the primed stem-like state may prevent neoplastic transformation in those systems (Lines: 413-414, 535-538). Nonetheless, our study highlights the need of testing the impact of fine-tuned Drp1 repression in stem cell maintenance and neoplastic transformation in other models. Please note, we refrain from drawing any conclusion about skin biology from this single cell line study.

2. Additional weaknesses include the reliance on a single carcinogen and on a single blot for expression of the fission and fusion proteins.(Editor's comment : likewise, repeating all experiments with a second carcinogen and blot will pose a big challenge. In consultation with the referees, addressing this particular concern is optional and left up to the author's discretion. However, do address the reason why this concern was raised (below in 8))

We thank the Editor and the Reviewers for understanding that repeating experiments with yet another carcinogen is a big challenge. We also reason this effort is unnecessary because we used the standard TCDD driven transformation of the skin keratinocyte line, HaCaT, as a model carcinogen to illustrate the main conclusions. Moreover, our finding that close to complete knockdown of Drp1 prevents TCDD driven neoplastic transformation is consistent with the findings from other labs in other cell models of Ras driven other neoplastic transformation (Lines: 413-414, 535-538). Nonetheless, our study highlights the need for testing if fine-tuned Drp1 repression accelerates neoplastic transformation by other carcinogens or specific driver mutations. Please note, we refrain from drawing any conclusion about the biology of TCDD from this single cell line study.

The relevant concern on immunoblot is addressed below in 8.

3. This is a concern because the increased levels of Drp1 in the TF1 cells in Figure 1D conflicts with what appears to be an overall decrease in Drp1 levels in TF1 in Figure 3C.

We thank the Reviewer for this comment. We have seen consistent increase in Drp1 protein levels in the TF-1 and TF-10 cells in comparison to the Parental, albeit to variable degrees. Now we have quantitated the presented blot after normalizing with the loading control, Actin, and the Parental population (Figure 1D and its legend). Now, we also provide another example of increase in Drp1 levels in TF 1 and TF10 while the increase in Mfn1 in the same samples is less consistent (Figure 1—figure supplement 1C and its legend, Lines: 130-132).

Clarification on Figure 3C of the first version of the manuscript:

Please note that the immunocytochemistry studies are less quantitative than immunoblotting (now clarified in Methods, Lines: 675-678). This is exemplified in the smaller increase in *Sox2* in the TF-1 cells in immunocytochemistry (Figure 3B,C,G) in comparison to the ~8 fold increase observed in immunoblot (Figure 3A). We noted immunocytochemistry with the mouse Drp1 antibody in use shows the expected elevation in Drp1 in the TF1 population, while the rabbit Drp1 antibody doesn’t (previous Figure 3C). Unfortunately, the mouse Drp1 antibody cannot be used to co-stain with mouse *Sox2* antibody in use. We have removed previous Figure3C, given comparison of Drp1 vs *Sox2* relationship between the cell populations may be misleading, although that within each cell population is accurate. Importantly, the inverse relationship of pDrp1-S616 vs *Sox2* is more robust. Moreover, we have now included data demonstrating the increased *Sox2* levels in the TF-1 population is maintained by reduced Drp1-S616 phosphorylation (Figure 3I and its legend, Lines: 329-356), along with other relevant data (Figure 3H, Figure 3—figure supplements 1B,C,D and their legend, Lines: 329-356).

4. The authors use an unusual method to characterize mitochondrial dynamics:Explain why other existing methods for quantification that give more meaningful numbers and are more readily understood by readers were not used.Is it also possible that a change in fission rates is balanced by a concurrent change in fusion rates, as suggested by the increase in Drp1 and Mfn1 levels in Figure 1C. The authors should address this, since so much of the paper rests on the assumption that fission rates are decreased in a subpopulation of cells.(Editor's comment: Address this in the text, no new experiments needed)

We thank the reviewer for this comment. Although deemed optional, we have added new data and analyses (including standard method, Figure 1—figure supplement 1D) to address this concern and add value to the revised manuscript (Lines: 134-165). Also, we have clarified the purpose and advantage of using the quantitative mitochondrial [Fission] and [Fusion] metrics without which we could not have identified the nuanced mitochondrial features of the stemness relevant subpopulations.

We assume the reviewer is alluding to the photoconversion based live cell pulse assay as the more readily understood method for mitochondrial dynamics. However, this valuable method for assessing overall mitochondrial dynamics and/or mitochondrial matrix continuity, cannot distinguish between ‘contribution’ of mitochondrial fission and fusion processes in maintenance of the steady state mitochondrial structure/morphology in single cells. To fill this gap, we previously designed, validated (using Drp1 and mitofusin knockouts) and demonstrated the utility of the mitochondrial [Fission] and [Fusion] metrics (as a part of the mitoSinCe^2^ assay and described in Methods, Lines: 600-604). Please note, as described in our original publication, mitochondrial [Fission] and [Fusion] metrics do not quantify kinetics of mitochondrial fission or fusion. Rather, these two quantitative metrics together can identify and quantitate heterogeneity in mitochondrial structure/shape in a given cell population. In our current work, use of these metrics provide the following quantitative advantages over the photoconversion based live cell pulse assay:

1) Validation of the metrics in Drp1 manipulated cells (Figure 3—figure supplement 1C, Figure 4I and their legends, Lines: 333-336, 427-431), and also revelation of nuanced differences with fine-tuned repression of Drp1 (Figure 1E, 4I and their legends, Lines: 153-165, Lines: 427-454), thus identifying the primed stem cell linked characteristic mitochondrial shape in individual cells.

2) Quantitation of Fusion1 (percent length of the longest mitochondrial element) revealed that fine-tuned reduction of Drp1 protein levels or its Drp1-S616 phosphorylation restricts the length of mitochondrial tubules in a network (Figure 1—figure supplement 1F, Figure 3—figure supplement 1C, Figure 4—figure supplements 1I,J,K, Figure 5F and their legends).

3) Validation of the mitochondrial mass alteration predicted by scRNAseq data (Figure 4J, 5B and their legends, Lines: 477-478).

4) Revelation of potential distinction between the mechanism of maintenance of mitochondrial shape in Parental vs its transformed TF-1 counterpart, which remains to be investigated further (Figure 3—figure supplement 1C, Lines: 333-342, 366-368).

Nonetheless, now we have added data using our refined photoconversion based matrix continuity assay (Lines: 134-165). This data confirms higher matrix continuity in both transformed cells in comparison to the non-transformed Parental cells (Figure 1—figure supplement 1D), as predicted by their ~10 fold less Drp1 activity (assessed by pDrp1-S616 / Drp1) (Figure 1D). As the reviewer suggested, this elevated matrix continuity likely results from unopposed fusion activity driven by mitofusins and Opa1 in the presence of repressed Drp1 activity. We found that the lower matrix continuity of TF-1 cells than TF-10 cells with 2 fold higher Drp1 activity is due to enrichment of a cellular subpopulation with moderately reduced [Fusion] and moderately elevated [Fission], as identified and quantified by the bivariate analyses of the metrics (Figure 1E). This stem cell linked subpopulation of cells is enriched when Drp1 is weakly repressed, whereas, more complete Drp1 repression expectedly increases the cell population with mitochondrial hyperfusion (Figure 4I and its legend; Lines: 427-449). Given Drp1-WT overexpression in the Parental cells increases [Fission] and reduces [Fusion] (Figure 3—figure supplement 1C and its legend) we conclude that the remnant Drp1 driven mitochondrial fission after fine-tuned Drp1 repression prevents mitochondrial hyperfusion to maintain smaller fused mitochondria (with reduced mitochondrial content), towards sustaining a [CyclinE^hi^-Sox2^hi^-Krt15^hi^] state (Lines: 449-460). Notably, Drp1 overexpression by itself does not support Sox2hi/Krt15hi state in Parental cells (Figure 3I) as does weaker repression of Drp1 (Figure 4A), suggesting Drp1 needs to be repressed to support stemness and further activated to support other cellular process.

Reviewer 3.1. Quantification of immunoblots in Figures 1D, 3E, 3F: Where the immunoblots are used to make arguments about increases and decreases in key proteins such as pDrp1 or Krt15 levels, these must be quantified in some manner.(Editor's comment : to avoid delays in publication, this is optional if it will require new experiments, but do quantitate existing blots if possible and address the issues raised)

We thank the reviewer for this comment. We have now quantitated all the previous and newly added blots (Figure 1D, 3A,D, 4A,E, Figure 1—figure supplements 1B, C). Key findings from immunoblots are consistent with data from single cell RNA-seq, immunofluorescence and microscopy analyses, as clarified in the manuscript (mentioned in relevant Result sections in the manuscript).

2. The placement of the arrow in 2B is quite confusing and doesn't seem to indicate the main point trying to be made which is more about Cluster 3 and Sox4 and Krt15 than Krt13 which doesn't seem to change.

We thank the reviewer for this comment. We have now removed the arrow and marked the slow cycling Krt15-high Cluster 3 by a box (Figure 2A, D).

3. Quantification of Dot plots: While the dot plots are helpful for seeing differences between individual cells within the populations, several conclusions are based on the interpretation of how many cells from one condition vs another are within a certain range or quadrant but these are not quantified. For example, 3B what percent of each condition are in each quadrant? This is hard to see by eye. Similarly, 3C, 3D, 3I, and 4B must be quantified and some measure of statistical significance presented. And what is the p-value shown in 4A referring to? Differences between all of them or individual comparisons?

We thank the reviewer for this comment. We have now quantitated all the dot plots (Figure B,C,G, and their legends), including the new data (Figure 1E and its legend). The quadrants boundaries have been determined by MAD (Mean Absolute Deviation), percent populations have been quantified for each relevant population, and statistical significance have been analyzed by Chi-Square test on all the populations compared within the black outline in each figure (Figure 1E, 3B,C,G, 4I). This is now described in Methods section (Lines: 655-658) and in the Figure legends, as appropriate.

4. The resolution of figures 4D and 4E must be improved so that the gene names can be seen upon zooming in

We thank the reviewer for this comment. We have now included a high resolution magnified figure in the supplementary (Figure 5—figure supplements 2A,B) where gene names are legible. Due to space constraint the figure couldn’t be enlarged in the main figure.

5. 4C: Is the difference in fusion1 statistically significant? how do the other populations look in the fission vs fusion1 plot?

We thank the reviewer for this comment. Figure 4C, now Figure 4K, is obtained by pooling all the cells within a population. We now added data demonstrating the same is true when Drp1 knockdown efficacy is lowered (Figure 4—figure supplement 1C and its legend, Lines: 440-445) and also for the transformed TF-1 and Tf-10 populations (Figure 1—figure supplement 1F and its legend, Lines: 161-164). Consistently, overexpression of less active Drp1-S616A mutant shows a similar phenotype but only in the TF-1 (Figure 3—figure supplement 1C and its legend, Lines: 333-340) (details in Response to point 9b-2). Also, we have included the scatter plot of Fission Vs Fusion1 for all other populations in Figure 4K (Figure 4—figure supplement 1B and its legend).

6. Ext Figure 2: 2a: may be better as a table instead of a graph because the graph is hard to read and compare. 2c-2d: Cluster 5 seems to be de-enriched for many genes so is it truly surprising or significant that there is less mtDNA? Are the overall counts lower? How are the data normalized to account for differences?

We thank the reviewer for the comment. Please note the Y axes in Figure 2—figure supplement 1A are same, which makes visual comparison between the populations easier than a table. Therefore, we now added the actual % numbers on top of the respective bar in the graph and mentioned this in the legend of the figure.

Indeed, Cluster 5 is characterized by overall reduction of various genes, where the mt-DNA genes are the most prominent ones as depicted by the color coded index (Lines: 213-215).

The gene expression is normalized with the standard NormalizeData function of Seurat (Methods, Lines: 647-648), where counts for each gene in each cell are divided by the total gene expression counts for that cell and multiplied by the scale.factor, 10,000. This is then natural-log transformed. Our preliminary data shows the reduction of mt-DNA genes happens after early exposure to 1 nM TCDD. We are currently investigating this in detail.

7. Extended Figure 4A (in figure legend or text please indicate the actual efficiency values and show error bars for replicates as the level of Drp1 repression/kd efficiency is a vitally important metric)

We thank the reviewer for the comment. We agree that the level of Drp1 repression/kd efficiency is very critical in supporting our main conclusion about the impact of fine-tuned Drp1 repression. Towards this end, we have provided the following revisions:

1. Added comparable metrics for Drp1 knockdown efficacy from immunoblots and scRNA-seq quantitation. We used the appropriately normalized Drp1 levels after knockdown that is further normalized by levels after control-knockdown (Lines: 463-467). We found that the knockdown difference between the shRNAs is 10 folds at the protein level (Figure 4A and its legend), and less than 2 fold at the RNA level (Figure 5—figure supplement 1A and its legend).

2. We performed fresh knockdown experiments further lowering Drp1 knockdown efficacy where both shRNAs reduced Drp1 levels by less than 2 folds at the protein level (Figure 4E and its legend; such remnant Drp1 levels may remain overestimated due to the reduction of the Actin control with Drp1 knockdown, Figure 4A,E). Nonetheless, such modest Drp1 knockdown reduces abundance of cells with >80 [Fusion1] (Figure 4—figure supplement 1C and its legend, Lines: 440-445), increases abundance of self-renewing cells and accelerates neoplastic transformation in Parental cells (Figure 4F, G and their legends, Lines: 402-407). Furthermore, plotting the accelerated transformation efficacy with Drp1 protein levels predicts ~50% repression of Drp1 protein levels can maintain maximal transformation efficacy within the experimental range (Figure 4—figure supplement 1A and its legend, Lines: 415-419).

3. Now we provide multiple analyses of the impact of overexpression of Drp1-wild type and the phospho-deficient Drp1-S616A mutant (Lines: 329-356). Our data suggest that elevated *Sox2*-hi/Krt15-hi sub-population is maintained by reducing Drp1-S616 phosphorylation of the elevated Drp1 protein levels in the TF-1 population, but not in the Parental (Figure 3I and its legend). Consistently, overexpression of Drp1-S616A mutant fails to cause the increase in the [Fusion] metrics observed with Drp1-WT overexpression only in the TF-1 (Figure 3—figure supplement 1C and its legend) (more details in Response to point 9b).

Therefore, our new data strengthens our conclusion that fine-tuned repression of Drp1 accelerates initiation of neoplastic transformation. Moreover, our new data also highlights that maintenance of the stem/progenitor cell state in the transformed TF-1 cells is achieved by fine tuning Drp1 activity by reducing its phosphorylation.

8. No key in extended Figure 4F.

We thank the reviewer for this comment. We have added the key in now Figure 5F.

9. Materials and methods: There are several key details missing such as the identity (manufacturer) and concentration of all of the antibodies used in the immunoblots and immunofluorescence analysis such as those for pDrp1, Opa1, Mfn1, Mfn2, Cyclins E2, A2, B1, D2, Cytochrome C, Fis1, Tom20. These details are not only important for reproducibility but as well as for evaluation of the results by experts familiar with these particular reagents.a. For immunoblotting of whole cell lysates, how did the authors make sure equivalent amounts of cells were analyzed, especially given their very different proliferation characteristics?b. How many replicates were used for the single cell RNASeq analysis of TF-1/TF-10 and of weak/strong Drp1-shRNA? What was the difference between the replicates especially for the cluster analysis as done in figure 2A (can this be shown in supplemental material?) to get an idea of how reproducible the findings are especially given how such a small difference in Drp1-shRNA induced levels seems to have such a striking difference on gene expression.c. Additionally, there are several typos in this section (p22 Lines 479, 481, 491. p24 lines 524, 525, 529).

We thank the reviewer for this comment. We have now added the details about the antibodies asked for in the Key Resources Table under Materials and methods.

For the immunoblots, we always load cell equivalents (assessed by counting a fraction of the cell suspension) and confirm with the loading control Actin or others, as appropriate. We have also verified by protein estimation, as appropriate. These details have been now included in the in the Methods (Lines: 577-581).

Please note, the 6 scRNA-seq based cell clusters defined by the top cluster markers identified in Figure 2A are also obtained by the second scRNA-seq data set done separately (Figure 5C and its legend, Lines: 488-493); this is irrespective of the drug treatment and selection of the second set. Furthermore, to confirm reproducibility of the overall cluster organization between the two experimental set, we identified all the common genes defining the clusters in the two experimental data sets (Figure 5—figure supplement 1E and its legend), albeit with some differences. Furthermore, the key findings of the scRNA-seq results are validated by immunoblots (*Sox2* and Krt15 levels in Figure 3A, 4A), immunocytochemistry (which is less quantitative, *Sox2* and Krt15 levels in Figure 3B) and quantitative microscopy analyses (mitochondrial mass in Figure 4J); all mentioned in the relevant Result sections.

Also, please note that the weaker Drp1 knockdown maintains 10 fold higher remnant Drp1 protein levels than the stronger Drp1 knockdown (Figure 4A and its legend). This 10 fold excess of Drp1 can potentially prevent the proposed compensatory adverse effect of excessive Drp1 suppression, namely increase in mitochondrial mass and cell cycle disturbances observed in our scRNA-seq data (Figure 5). Moreover, we predict this compensatory effect may be detected beyond ~50% repression of Drp1 protein when transformation efficacy starts declining (Figure 4—figure supplement 1A) (details in point 16) (Lines: 483-487).

We have corrected all the typos in the Materials and methods section, and others.